# Human brain changes after first psilocybin use

T. Lyons[1], M. Spriggs[1], L. Kerkelä[2], F. E. Rosas [1,3,4,5,6], L. Roseman [1], P. A. M. Mediano [5,7,8], C. Timmermann [1], L. Oestreich[9], B. A. Pagni [10], R. J. Zeifman[1,10,11], A. Hampshire[12], W. Trender [13], H. M. Douglass [1], M. Girn [14,15], K. Godfrey[1], H. Kettner[1,15], F. Sharif [1], L. Espasiano[1], A. Gazzaley [16], M. B. Wall [17], D. Erritzoe [1], D. J. Nutt [1] & R. L. Carhart-Harris [1,15] ✉

Psychedelics have robust effects on acute brain function and long-term behavior but whether they also cause enduring functional and anatomical brain changes is largely unknown. In an exploratory, placebo-controlled, within-subjects, electroencephalography (EEG), and magnetic resonance imaging (MRI) study in 28 healthy, entirely psychedelic-naive participants, anatomical and functional brain changes are detected from one-hour to one-month after a single high-dose (25 mg) of psilocybin. Increases in cognitive flexibility, psychological insight, and well-being are seen at one-month. Diffusion tensor imaging (DTI) done before and one-month after 25 mg psilocybin reveals decreased axial diffusivity bilaterally in prefrontal-subcortical tracts that correlate with decreases in brain network modularity (fMRI) over the same month. Enduring functional brain changes are largely absent, but network modularity change (numerical decrease) negatively correlates with well-being change (significant increase), in line with previous findings in depression. Increased cortical signal entropy (EEG) at 1- and 2-hours post-dosing predicts improved psychological well-being at one-month. Next-day psychological insight mediates the entropy to well-being relationship. All effects are exclusive to 25 mg psilocybin; no effects occur with a 1 mg psilocybin placebo.

Psilocybin (*4-phosphoryloxy-N,N-dimethyltryptamine*) is the precursor of psilocin (*4-hydroxy-N,N-dimethyltryptamine*), a serotonin receptor agonist. Converging evidence supports a role for serotonin 2A receptor (5-HT2AR) agonism in eliciting the characteristic brain and subjective effects of this and related psychedelics in humans[1,2]. Preclinical (i.e., mostly mouse) research supports an association between 5-HT2AR agonism and increased markers of anatomical neuroplasticity, such as synaptogenesis[3–7]. One study in mice[8] challenged the selective importance of 5-HT2AR signaling for inducing synaptogenesis, but used a high dose of a 5-HT2AR antagonist that likely caused off-target and therefore non-5-HT2AR-specific effects.

Clinical trials have assessed psilocybin-therapy for wide-ranging psychopathology[9–14]. Human research has found enduring psychological changes after a single high-dose psilocybin[15], including increased cognitive flexibility in depressed patients[16] (although see ref. [17] with LSD in healthy volunteers) and improved well-being in diverse samples[18,19]. Several human neuroimaging studies have observed changes in acute brain function with psychedelics[20–25]. Long-term brain changes are less well characterized but see refs. [26,27] in healthy volunteers and [16,28–31] in depression. See Table S1 for a list of relevant studies.

In the present work, we sought to address important knowledge gaps regarding human brain changes with psilocybin, such as the so-called 'entropic brain effect'[32,33] and whether it can predict salient psychological outcomes, plus whether enduring functional or anatomical brain changes occur after someone's first psychedelic experience. Eyes-closed, task-free electroencephalography (EEG), functional

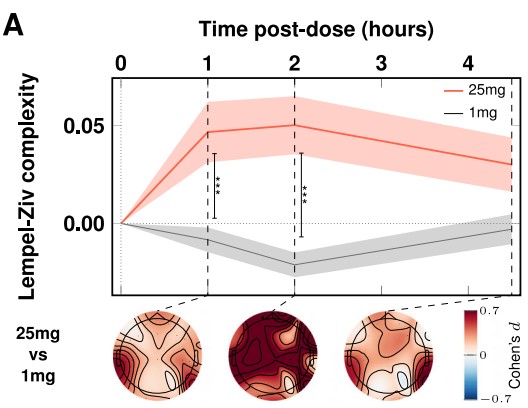
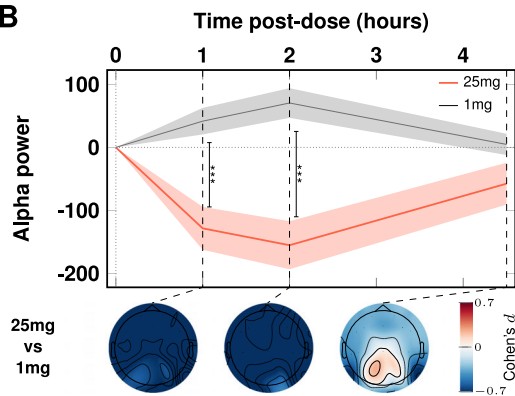
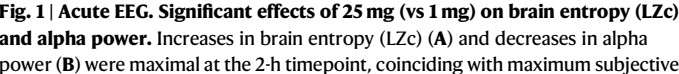

**Fig. 1 | Acute EEG. Significant effects of 25 mg (vs 1 mg) on brain entropy (LZc) and alpha power.** Increases in brain entropy (LZc) (**A**) and decreases in alpha power (**B**) were maximal at the 2-h timepoint, coinciding with maximum subjective intensity. The red (25 mg) and gray (1 mg, placebo) timelines correspond to means and standard errors (shaded areas) at 1, 2, and 4.5 h post-dose. Vertical bars index contrasts, *** = $p < 0.001$. See Fig. S3 for single-subject data.

magnetic resonance imaging (fMRI) and diffusion tensor imaging (DTI) recordings were done in healthy human volunteers during (EEG), as well as before and (1-month) after (fMRI, DTI) they received their first-ever high-dose of a psychedelic (25 mg psilocybin). Brain (fMRI/DTI) and behavioral outcomes were assessed at baseline and 1-month post-dosing. EEG was done once prior to dosing and three times during dosing (1, 2, and 4.5 h). Due to prior evidence of enduring behavioral changes after psilocybin[16,27,28,30], a fixed-order, repeated measures design was used; participants received an initial "placebo" dose of psilocybin (1 mg) 1-month prior to a subsequent 25 mg dose. Dosing was oral. This was an exploratory, hypothesis-generating mechanistic study in healthy volunteers. It was therefore analyzed accordingly. See section 2.1. of the supplement, and Fig. S1 and Table S2, for details of the study design.

## Results

### Demographic data
Twenty-eight healthy volunteers participated, mean age = 41 (SD = 8.7), 43% female and 57% male. See Table S3 for demographics.

### Acute brain effects: time-dependent changes in brain entropy; Lempel-Ziv complexity, (LZc), and spectral power (EEG)
Eyes-closed, task-free 'resting-state' brain activity during each of the two psilocybin dosing sessions (i.e., 1 mg and 25 mg, 1-month apart) was investigated using EEG recorded at pre-dose baseline and 1-, 2-, and 4.5-h post-dosing. Large pre vs acute (i.e., 'on-drug') brain changes were observed after the 25 mg dose, as revealed by *Dose* (1 mg vs 25 mg) × *Time* (0-4.5 h) linear mixed-effects modeling. There were significant ($p < 0.001$) increases in the informational entropy of spontaneous scalp potentials (LZc) at 1- and 2-h post 25 mg, and significant decreases in alpha power (Fig. 1A, Fig. 1B). LZc indexes the statistical irregularity or incompressibility of a timeseries, hence its information-theoretic *entropy*. Additionally, power increases in the gamma band at 2 h and decreases in the theta band at 1 and 2 h were observed. No changes in EEG recorded activity were observed under 1 mg psilocybin (See Table S4 and Fig. S2 for these data).

### Long-term brain changes: anatomical changes (MRI)
Diffusion tensor imaging revealed a significant interaction between 'timepoint' (i.e., one factor, three-levels, 1) baseline, 2) 1-month post-1mg, 3) 1-month post-25mg) and mean axial diffusivity (AD)—the diffusion coefficient along the principal diffusion direction, see Fig. 2A. ANOVA revealed significant changes in AD for two tracts after 25 mg psilocybin, a bilateral prefrontal cortex - striatum tract (PFC-STR; $F(2, 48) = 10.30$, $p = 0.005$) and a PFC–thalamus tract (PFC-THA; $F(2, 48) = 9.51$, $p = 0.008$). See Fig. S13 for all relevant tracts. Post-hoc t-tests

found significant decreases in AD 1-month post-25mg vs. 1-month post-1mg for the PFC-STR ($t(24) = -3.72$, $p = 0.006$) and PFC-THA tracts ($t(24) = -3.85$, $p = 0.005$). It should be noted that the PFC-STR and PFC-THA tracts show substantial spatial overlap, suggesting that the AD changes in these two tracts are related and do not represent independent findings. This convergence arguably supports the interpretation that the change has a (common) PFC origin. Separating by hemispheres for post-hoc t-tests, decreases were apparent in both hemispheres; however, with Bonferroni correction, only decreases in the left hemisphere remained significant (see Fig. S14). No significant differences in AD were observed following the 1 mg control. With free-water correction (see Fig. S16 and section 5.5 of the supplement), consistent AD effects were seen post-25mg psilocybin, and fractional anisotropy (FA) changes also became statistically significant (Fig. S16). A tract-based spatial statistics (TBSS) method was also done on request (see "Methods"), but yielded no significant results.

### Long-term brain changes: brain response to emotional faces
Whole-brain analyses of the Blood Oxygen Level Dependent (BOLD) fMRI responses to emotional face stimuli (happy, fearful, neutral) yielded a large sized effect (95% CI [0.33, 1.34], $d = 0.9$) for the salient contrast of one-month post-25mg ($t(24) = -3.85$, $p = 0.005$) vs 1-month post-1mg psilocybin for fearful faces vs fixation cross, Fig. 2C and 2D (see Section 4 of the supplementary file, Figs. S4 and S5, for results for happy and neutral faces—and for the post-25mg vs pre-dose baseline contrast; 95% CI [0.04, 0.98], $d = 0.3$). However, an all factors, all levels repeated-measures ANOVA (whole-brain), failed to find a significant interaction between emotional face type and *timepoint* on BOLD response. Explorative amygdala region of interest (ROI) results can be found in the supplement (Figs. S6 and S7), along with generalized psychophysiological interaction (gPPI) functional connectivity (FC) findings. Amygdala effects were observed for the left hemisphere and gPPI effects were observed that survived FDR correction and Bonferroni correction was done for exploring both hemispheres (See Table S5 and Fig. S8).

### Long-term brain changes: resting-state FC (RSFC)
Exploratory resting-state (RS) functional connectivity (RSFC) analyses were applied to data from a single 8-min eyes-closed, task-free, RS fMRI run. Four regions were initially chosen for seed-based RSFC analyses: (i) bilateral parahippocampus (PH), (ii) bilateral amygdala, (iii) ventromedial prefrontal cortex (vmPFC), and (iv) subgenual anterior cingulate cortex (sgACC). Two further seeds were examined after peer review: (v) the hippocampus and (vi) the dorsal anterior cingulate cortex (dACC). In ANOVAs across all timepoints, three of the initial four seed did not yield significant results and neither did further two seeds

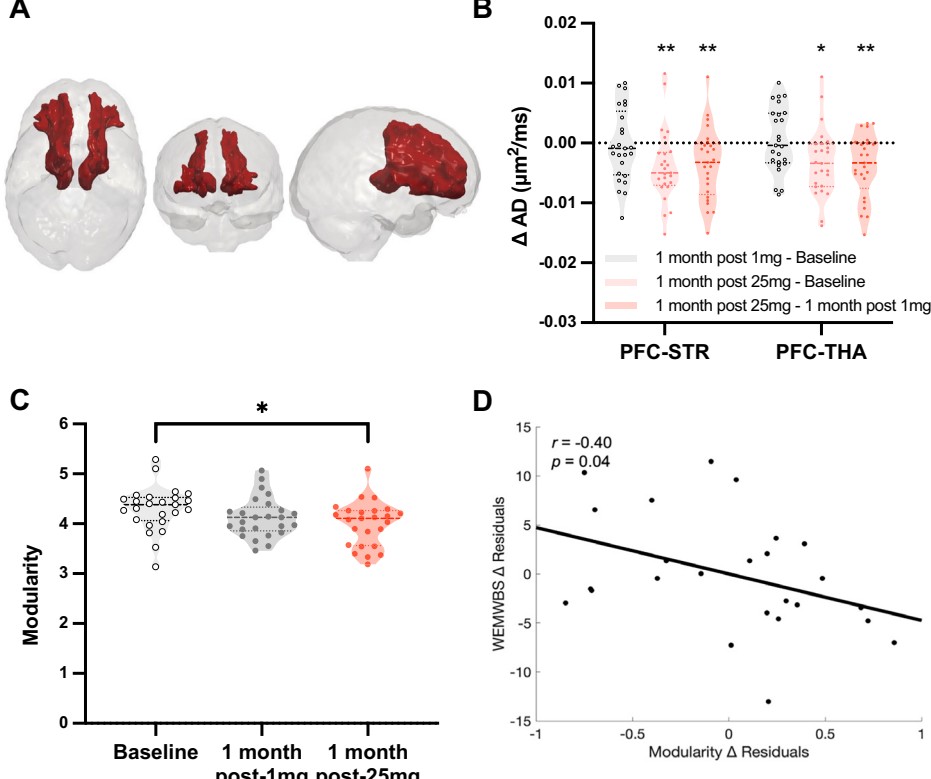

**Fig. 2 | One-month post-25 mg psilocybin anatomical and functional brain changes. A** DTI. Combined bilateral prefrontal-subcortical white-matter tracts where significant decreases in AD were observed 1-month post-25mg, Bonferroni corrected. **B** Violin plot shows decreases in AD with dose at one-month post-25mg vs baseline and vs key contrast one-month post-1mg (PFC-STR: $p = 0.006$; PFC-THA: $p = 0.005$); * $= p < 0.05$, ** $= p < 0.01$. See supplement for consistent results with free-water correction (Fig. S16) as well as single-subject data (Fig. S15). **C** *Brain network modularity* at the three timepoints. A significant decrease was observed 1-month post-25mg vs baseline ($p = 0.02$); * $= p < 0.05$. **D** Controlling for *well-being* prior to 25 mg, there was a significant negative correlation between *modularity change* and *well-being change* in the key contrast of one-month post-25mg vs 1-month post-1mg ($r = -0.4$, $p = 0.04$, 2-tailed), i.e., decreased modularity correlated with improved

well-being (see also Fig. 3B). Note that the residuals are plotted in Fig. 2D. Plotting the residuals does not highlight the substantial size of the improvements in well-being that were seen (note, these changes *are*, however, visible in 4C). Effects were tested using two-tailed linear mixed-effects models and Bonferroni correction for multiple comparisons. For further DTI and fMRI results, including single-subject data, see the supplementary file. Modularity and AD (PFC-STR and PFC-THA tracts merged) change values for the salient one-month post-25mg vs one-month post-1mg contrast also contribute to Fig. 3B. AD axial diffusivity, prefrontal cortex - striatum tract (PFC-STR), prefrontal cortex - thalamus tract (PFC-THA), RSFC 'resting-state' functional connectivity, WEMWBS Warwick-Edinburgh Mental Well-being Scale.

that were assessed on request in review. See Fig. S9 for the relevant maps and Fig. S10 for single-subject data.

## Long-term brain changes: network modularity (RSFC)

Previous work highlighted a relationship between improved depressive symptoms and decreased brain network modularity after psilocybin-therapy for depression[28]. Brain network modularity computes the extent to which brain functional connectivity can be divided into distinct networks or modules of brain regions that are more strongly connected within a network than to other regions outside of the network. If brain network modularity decreases, it implies a less segregated−or more globally integrated−profile. Unlike in previous psilocybin for depression work where post-treatment decreases in modularity were observed, statistically significant group-average decreases in modularity were not observed one-month post-25 mg vs. 1-month post-1mg ($p = 0.209$; $d = 0.3$) psilocybin, yet a statistically significant decrease in modularity was observed when post-25mg was contrasted against pre-dose baseline ($t(24) = -2.95$, $p = 0.007$; $d = 0.6$). No changes in mass univariate network analyses were observed. We found no statistically significant changes in *within* - or *between* network RSFC at 1-month post-25 mg vs. 1-month post-1 mg, and no differences in any of the above-listed analyses were observed at 1-month post-1mg vs pre-dose baseline.

Controlling for *well-being* before the 25 mg session, a correlation was found between *modularity* changes (decreases) 1-month post-25 mg and contemporaneous changes (improvements) in *well-being* ($r = -0.40$, $p = 0.04$, two-tailed; Fig. 2D and also Figs. 3B and S12). This relationship is consistent with prior work that found decreased network modularity post-psilocybin correlating with improved depressive symptom severity in two separate psilocybin-therapy for depression trials[28].

## Psychological outcomes: acute subjective intensity

An interaction between *dose* (1 mg vs 25 mg) and *time* (0–6 h) on the *intensity* of subjective drug effects was seen ($F_{(8,197)} = 57.73$, $p < 0.0001$). Greater subjective intensity was reported in relation to 25 mg psilocybin compared with both time-zero baseline (all $p < 0.0001$ from 1 to 5 h; $p < 0.001$ at 6 h) and the 1 mg control (all $p < 0.0001$ from 1 to 5 h; $p = 0.001$ at 6 h; Fig. 4A). No significant elevation in subjective intensity was observed with 1 mg. All, except one, of the participants (94%) rated the 25 mg experience as the "*single most unusual state of consciousness*" of their entire life. The remaining person ranked it "*among the top five most unusual experiences*" of their entire life. In contrast, most participants rated their 1 mg experience as "*no more unusual than an everyday state of consciousness*" (see Fig. S20) reinforcing the impression−from the EEG data−that the 1 mg 'placebo' dose was functionally inactive.

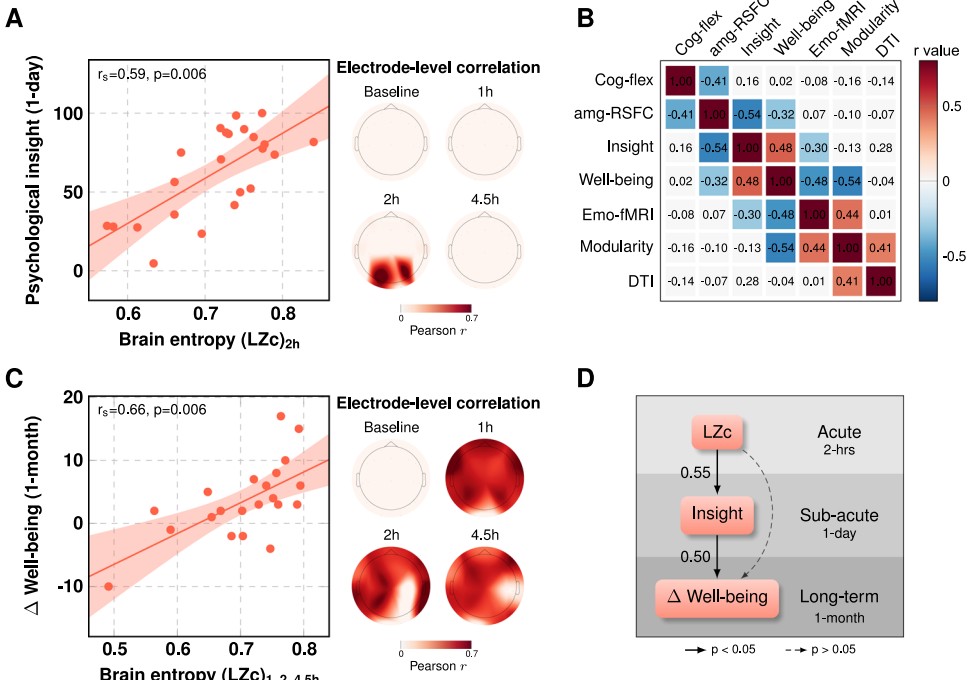

**Fig. 3 | Correlational and predictive modeling. A** A cross-validated predictive model revealed a significant positive correlation between acute LZc (or 'brain entropy') for P3, P4, and O1 electrodes, and psychological insight, measured 1-day post-25mg. Inset shows electrode-level relationship between LZc (brain entropy) at 2 h post-dose (when subjective effects were intense) and next-day insight (see also 3.4 in the supplement for a validation of this result using data from all sensors). Data are expressed as mean ± SEM. **B** Correlation matrix examining relationships between functional and anatomical MRI metric values and contemporaneous psychological outcomes. Colored cells represent significant Pearson correlations, corrected for multiple comparisons (non-parametric cluster test, p = 0.006). Red/pink cells reflect significant positive correlations; blue cells reflect significant negative correlations. All relationships are directionally consistent with the original direction of changes, suggesting a coherent factor of change. All values derive from the key one-month post-1mg versus one-month post-25mg contrast. Cluster test is correcting for 21 tests. **C** Model predicting one-month well-being increases directly from acute increases in brain entropy at all three post-dosing timepoints. Plot shows electrode-level relationship with (average) data from the 3 post-dose timepoints, i.e., 1-, 2-, and 4.5-h post-dose (see also 3.4 in the supplement for a validation of the result using data from all sensors). Data are expressed as mean ± SEM. **D** Model shows that increased well-being at one-month post-25 mg can be explained by increased brain entropy (LZc) across all sensors, but also that the strongest path entails *brain entropy* first predicting next-day *insight*, which then, in-turn, predicts improvements in *well-being* (1-month). The values between the arrows in Fig. 4D represent normalized regression coefficients. The relationships were statistically significant at $p < 0.05$. amg-RSFC amygdala-resting state functional connectivity; Cog-fle Extradimensional shift *errors*; DTI diffusion tensor imaging; emo-FMRI *BOLD responses to emotional faces*; LZc Lempel-Ziv complexity.

## Psychological outcomes: psychological insight

Psychological insight was assessed via the validated 'psychological insight scale' (PIS)[34]. Analyses revealed a significant interaction between *dose* (1 mg vs 25 mg) and sub-acute *timepoints* on *insight* ($F_{(2,44)} = 14.05$, $p < 0.0001$; Fig. 4B). Insight scores were significantly higher after 25 mg versus 1 mg psilocybin at all sub-acute timepoints −i.e., one-day ($M_{diff} = 50.90$, $SE = 5.51$, $p < 0.0001$; 95% CI [39.50, 62.29], $d = 1.9$), 2-weeks ($M_{diff} = 29.83$, $SE = 4.72$, $p < 0.0001$; 95% CI [20.08, 39.57], $d = 1.3$) and one-month post-dosing ($M_{diff} = 27.86$, $SE = 4.37$, $p < 0.0001$; 95% CI [18.90, 36.82], $d = 1.2$), Bonferroni-corrected.

## Psychological outcomes: psychological well-being

Psychological well-being was assessed using the Warwick-Edinburgh Mental Wellbeing Scale (WEMWBS), a validated, single dimension, self-report measure of well-being rated with reference to a two-week period[35]. Analyses revealed a significant interaction between *dose* (1 mg vs 25 mg) and sub-acute *timepoints* on *well-being* ($F_{(2,44)} = 11.92$, $p < 0.0001$). Bonferroni-corrected post-hoc comparisons revealed significantly greater increases in well-being at two-weeks ($M_{diff} = 5.83$, $SE = 1.54$, $p < 0.01$; 95% CI [−10.64, −1.02], $d = 0.8$) and one-month ($M_{diff} = 4.70$, $SE = 1.47$, $p < 0.05$; 95% CI [−9.29, −0.10], $d = 0.6$) post-25mg psilocybin (vs 1 mg), Fig. 4C. There were no significant changes post-1mg dose vs (pre any intervention) baseline.

## Psychological outcomes: cognitive flexibility

Cognitive flexibility was measured via an intra-dimensional/extra-dimensional (IDED) task, optimized for internet-based delivery[36,37]. Extradimensional shift (EDS) is an index of participants' ability to correctly identify rule shifts, and thus, *cognitive flexibility*. Participants completed the task at baseline and 1-month post-dosing. Analyses revealed a significant interaction between *task discrimination stage* and *timepoint* on *cognitive flexibility* ($F_{(7.29,167.67)} = 2.47$, $p = 0.018$, $\varepsilon = 0.456$). FDR-correction of post-hoc comparisons (i.e., for the 9 IDED task phases, Figure S17) plus Bonferroni correction within the salient EDS phase, showed a significant decrease in EDS errors 1-month post-25mg vs 1-month post-1mg (Fig. 4D), indicative of greater cognitive flexibility after the 25 mg dose ($M_{diff} = 0.06$, $SE = 0.02$, $p = 0.016$; 95% CI [0.00, 0.12], $d = 0.5$). Contextualized by no global difference in performance or at IDS, IDR or EDR stages, the EDS change pertains to a flexible attentional set as opposed to perseveration or a more general problem with processing discriminations/maintaining rules. No IDED changes were observed after 1 mg psilocybin. See Figures S17-S19 and Table S6 of the supplement for more detailed IDED results.

## Correlations between long-term change outcomes

The relationship between various temporally-synchronized changes in brain and behavioral variables was assessed via a correlation matrix that included the relevant contrast values for: 1) *EDS errors* (Cog-flex);

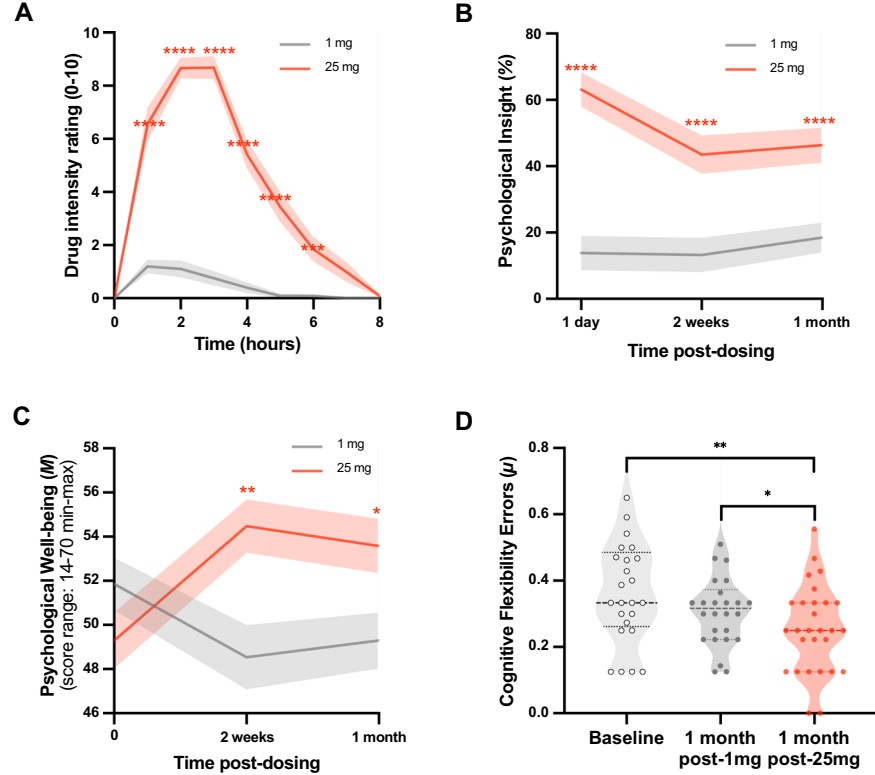

**Fig. 4 | Behavioral analyses. A Subjective intensity**. A significant interaction between *dose* (1 mg vs 25 mg) and *time* (0–6 h) ($F_{(8,197)} = 57.73$, $p < 0.0001$) on *subjective intensity* rated in-session. Post-hoc t-tests, 25 mg vs 1 mg, *** = $p < 0.001$, **** = $p < 0.0001$. **B Psychological insight scale (PIS)**. A significant interaction between *dose* (1 mg vs 25 mg) and *timepoint* (1-day, 2-weeks and 1-month post-dosing) on *psychological insight* ($F_{(2,44)} = 14.05$, $p < 0.0001$); insight scores (PIS) were significantly greater at all timepoints post-25mg vs 1 mg (**** = $p < 0.0001$). **C Warwick-Edinburgh Mental Well-Being Scale (WEMWBS)**. Potential scores on the WEMWBS range from 16 to 80. A significant interaction was observed between *dose* (1 mg vs 25 mg) and *timepoint* (pre-dose baseline, two-weeks & one-month post-dose) on *well-being* ($F_{(2,44)} = 11.92$, $p < 0.0001$); *well-being* (WEMWBS) increases were significant at two-weeks ($M_{diff} = 5.83$, $SE = 1.54$, $p < 0.01$; 95% $CI$ [−10.64, −1.02], $d = 0.8$) and one-month ($M_{diff} = 4.70$, $SE = 1.47$, $p < 0.05$; 95% $CI$ [−9.29,

−0.10], $d = 0.6$) post-25 mg psilocybin. Post-hoc *t*-tests, 25 mg vs 1 mg, * = $p < 0.05$, ** = $p < 0.01$. **D Intra- and extra-dimensional attentional set-shifting (IDED) and Extradimensional Shift Errors (EDS, *cognitive flexibility*)**. Analyses revealed a significant interaction between *task performance* and *timepoint* ($F_{(7.29,167.67)} = 2.47$, $p = 0.018$; see S18A). Post-hoc tests showed significantly fewer EDS errors in the key contrast of 1-month post-25 mg vs one-month post-1mg ($M_{diff} = 0.06$, $SE = 0.02$, $p = 0.016*$; 95% CI [0.00, 0.12], $d = 0.6$), outlier corrected, see Figure S18C for details. This suggests an effect of 25 mg psilocybin rather than a mere order (e.g., practice) confound. Fewer EDS errors reflects faster (correct) detection of rule shifts. Data are expressed as mean ± SEM. Effects were tested using two-tailed linear mixed-effects models and Bonferroni correction for multiple comparisons. T-tests, * = $p < 0.05$, ** = $p < 0.01$. No significant effects of 1 mg were observed on any of the statistical tests.

---

2) *amygdala-RSFC* (amg-RSFC); 3) *well-being*; 4) *BOLD responses to emotional faces* (Emo-fMRI); 5) *brain network modularity*; 6) *DTI-measured axial diffusivity* in the merged PFC-THA and PFC-STR tracts (DTI), and 7) *psychological insight* (one-month post 1 mg vs 1-month post 25 mg); Fig. 3B. All correlations pertain to the key contrast of one-month post-1mg vs 1-month post-25mg. Values represent Pearson correlation coefficients, calculated via pairwise linear regressions. Significance was assessed via non-parametric cluster statistics, appropriate for non-independent tests. A significant cluster was found ($p = 0.006$, cluster corrected, colored cells), implying that the correlational structure was not due to chance. Each of the significant relationships is directionally consistent with the original direction of change in each outcome, again, for the key contrast of one-month post-1mg vs one-month post-25 mg psilocybin.

### Acute brain entropy predicts increased psychological insight and improved well-being

In two analyses, the informational entropy of spontaneous scalp potentials (*LZc*, EEG) under 25 mg psilocybin predicted both next-day psychological insight (Fig. 3A) and improvements in psychological well-being at one-month post-25mg psilocybin (Fig. 3C). In the former case, LZc increases within posterior electrodes (P3, P4, and O1) recorded during 2-h post-25mg, predicted psychological insight

scored one-day later ($r_s = 0.59$, $p = 0.006$, Fig. 3A). In the latter case, LZc predicted improved psychological well-being scores (one-month post-25mg versus one-month post-1mg) directly from most sensors at all three post-dose timepoints (i.e., at 1, 2, and 4.5 h; $r_s = 0.66$, $p = 0.006$, Fig. 3C). Note that regressions were done via cross-validation—where the most sensitive sensors for the relationship—i.e., those with the greatest predictive capacity—were identified. However, repeating these analyses using all sensors produced consistent results (see 3.4. in the supplement). No predictive relationships were found (in any of the sensors) for the 1 mg condition.

Finally, a mediation analysis (Fig. 3D) was performed using linear models that included key variables at three timepoints: (i) whole-brain LZc 2 h post-25mg psilocybin (i.e., averaged signal across all sensors at this timepoint), (ii) *insight* measured 1-day post-25mg, and (iii) *well-being* changes 1-month post-25mg. This analysis revealed that acute brain entropy (*LZc*) was a significant predictor of next-day *insight* ($\beta = 197$, $SE = 67$, $p = 0.008$, normalized $\beta = 0.55$). Both *LZc* ($\beta = 41.8$, $SE = 14.2$, $p = 0.008$, normalized $\beta = 0.54$) and *insight* ($\beta = 0.11$, $SE = 0.04$, $p = 0.01$, normalized $\beta = 0.50$) were significant predictors of improved *well-being* at 1-month. Importantly, satisfying the criteria for statistical mediation[38]—as well as more recent recommendations for defining this[39]—the overall predictive strength of the model was greatest if psychological insight was included. Adding *insight* into the

model weakened the direct connection between *LZc* and *well-being*; validating its role as a mediator of the *LZc* to *well-being* relationship.

## Discussion

This was an exploratory, single-blind, placebo-controlled, fixed-order, repeated measures multimodal neuroimaging study with mechanistic aims in healthy volunteers receiving their first ever psychoactive dose of a psychedelic— a high dose (25 mg) of psilocybin. EEG was recorded during dosing sessions and fMRI and DTI data were collected prior to and 1-month after each dosing session. Changes in properties of the BOLD fMRI signal were the primary outcome, yet effects were largely weak and results, non-significant. Conversely, robust increases in signal complexity—otherwise known as 'brain entropy'—were observed under 25 mg psilocybin that predicted salient psychological outcomes, such as psychological insight and improved well-being. DTI also yielded significant findings, with an apparent compacting—or thinning—of white matters fibers spanning the prefrontal cortex and subcortical regions.

The present work sheds light on human brain changes under and after first-time high-dose psilocybin. The high-dose session was each person's first-ever psychedelic experience. All—except one participant—rated their high-dose experience with psilocybin as the *single most unusual conscious state of their entire lives*. The one person who did not, ranked it within their *top-five most unusual conscious experiences*.

The multimodal neuroimaging design allowed us to observe changes in brain function and (potential) anatomy from 1-h (EEG) to 1-month (DTI) after high-dose psilocybin. No acute or long-term changes were apparent after a 1 mg placebo dose of psilocybin. Empirical modeling highlighted the predictive power of increased brain entropy under high-dose psilocybin, and validated its correlative relationship with psilocybin's characteristic "psychedelic" effects. A predictive relationship was also found between brain entropy and longer-term mental-health changes—namely, improved well-being. Improved well-being could be predicted directly from acute increases in brain entropy as early as 1-h post dosing. Prediction could also be done through a two-step path or sequence where increased brain entropy first predicted next day psychological insight, which then predicted the one-month improvements in well-being. If non-pharmacological variables are standardized—for example, if they are consistently 'psychologically supportive'[40], then these results imply that human brain changes as early as 1-h into a 25 mg psilocybin experience—and that seem closely related to the subjective psychedelic experience—can predict mental health improvements 1-month later. This finding lends further support[41] to the position that the psychedelic experience is involved in the therapeutic effects of psychedelic compounds. In terms of subjective effects, the present work only reports on generic intensity. Future work could examine additional, more specific dimensions of experience, such as emotional breakthrough[42].

The inclusion of DTI enabled us to test for long-term changes in the integrity of white matter tracts post psilocybin. Results revealed decreased axial diffusivity in prefrontal-subcortical tracts 1-month post 25 mg psilocybin. If verified in subsequent work, this finding may be viewed as evidence of anatomical 'neuroplasticity' after first-ever psychedelic use in humans, echoing earlier in-vivo animal studies following single-dose psilocybin that reported increases in synaptic spine formation in female mice[5] and synaptic density in pigs[6]. However, interpretation of the axial diffusivity changes is complex, not least due to crossing fiber-related confounds[43]. The diffusion-weighted signal can change due to neurofibril or glial growth, altered myelination, axon density, membrane permeability or extracellular fluid[44]. Decreases in axial diffusivity have been observed with meditation[45], healthy neurodevelopment[46,47] and learning[48,49], but also with axonal injury[50,51], ageing and related pathology[52].

5-HT2AR agonism has been associated with in vivo dendro-architectural changes in adult mouse brain[5], axonal development in embryonic mouse brain[53], and oligodendrocyte changes in rodent brain tissue in vitro[54]. The high density of 5-HT2ARs in the human prefrontal cortex[55] and greater number of cortical compared with subcortical terminals[56] suggest a PFC 5-HT2A receptor locus of action for the observed diffusivity changes. The DTI results were robust to free-water correction, and the AD decreases correlated with decreased brain network modularity at one-month post-25mg psilocybin, lending tentative support to the inference that functionally relevant microstructural changes had occurred. Further research using multi-shell sequences is needed to disambiguate the current findings and inform on their robustness and replicability. This work is currently underway. Until there is replication, caution is advised when interpreting the biological basis of the DTI findings reported here. If the effects *are* truly reflective of microstructural change, then decreases in AD and FA exclusively after the 25 mg dose might tentatively relate to two types of change: the pruning of weak or redundant connections and/or neurogenesis with under-myelinated axons.

Decreased EEG alpha power, a well-replicated effect of psychedelics, is linked with cortical disinhibition[57] and increased brain entropy under psychedelics appears to be dose-dependent[58] and has been associated with acute and sub-acute psychological changes[59,60]. Decreased cortical alpha power and increased brain signal entropy are robust and reliable markers of the acute action of psychedelics in humans[20,21,60–62]. Future research in larger samples could test whether acute EEG changes, induced by 5-HT2AR agonist psychedelics, can predict downstream anatomical neuroplasticity[3] or therapeutically-relevant functional brain changes[28]. However, we failed to find a relationship between acute increases in entropic brain activity and the downstream DTI changes here.

As reported above, compared with other outcomes in this study, the fMRI-measured brain changes assessed 1-month post 25 mg were relatively modest; for example, exploratory ROI and network-based RSFC analyses, including brain network modularity, yielded largely non-significant results, and a brief (8 min) emotional faces paradigm also provided little in the way of statistically significant results. The relevant details can be viewed in the methods section and supplementary file (Figs. S4–12). Previous work assessing post psilocybin therapy changes in brain function in individuals with depression have revealed more robust changes[16,28–30]. For example, in two separate trials of psilocybin for depression, decreased brain network modularity was seen that correlated with or predicted improvements in symptom severity post-treatment[28] (see also ref. [63]). Doss et al. found similar effects with psilocybin-therapy for depression that they labeled increased "neural flexibility"[16]. In the present study, changes in brain network modularity failed to reach statistical significance (Fig. 2), but we did see a *network modularity change* vs *mental health change* relationship (Fig. 2D) that was directionally consistent with previous work, i.e., decreased modularity correlating with improved mental health[28].

The present study's fMRI results suggest that enduring functional brain changes post psilocybin are less robust and reliable in healthy versus mentally unwell populations e.g., ref. [28]. If this is more generally the case, it could imply that greater baseline atypicality in clinical populations (i.e., baseline values being far from a healthy population average), biases a remediating change via psilocybin. However, given the reliability of enduring psychological changes post psilocybin in healthy samples[18,19], it remains plausible—if not likely—that functional brain changes do occur, but that their detection is dependent on—or sensitive to— experimental modalities, metrics and parameter decisions[28,30,64]. In short, we may not have yet discovered a sufficiently sensitive assay for detecting (true) functional brain changes post psilocybin. Further work is required to address this ongoing knowledge gap.

The present multi-modal neuroimaging study in healthy participants sheds light on the brain effects of first-time high-dose

psychedelic use and the therapeutic action of psilocybin-therapy, suggesting that therapeutically relevant changes—i.e., improved well-being—can be forecast via an *acute* human brain action, i.e., an entropic brain effect, that is well-known to relate to the psychedelic experience[60]. Recent evidence suggests that this characteristic effect of psychedelics is dose-dependent and somewhat exclusive to this category of drug versus psychoactive stimulants[25] and cannabinoids[58]. See here for a relevant review[33]. Long-term improvements in well-being were predicted by acute increases in brain entropy, mirroring the dynamic profile of psilocybin's acute subjective effects. Results support a role for psychological insight in mediating the causal association between the entropic brain effect and potentially enduring improvements in well-being. These and prior results highlight psychological insight[34] and entropic brain effects[32,65] as important for the therapeutic action of psychedelic-therapy. Possible white-matter changes, as well as improvements in well-being and cognition were also observed in this study. All acute and enduring psychological and neurobiological effects were exclusive to the high-dose psilocybin condition. By magnitude, the relevant post high-dose changes far-surpassed any potential order effects associated with the baseline versus placebo dose contrast.

Justified by prior evidence of enduring psychological changes post-psychedelics[9,10], this study used a fixed-order cross-over design. Pairwise contrasts on all the study's main outcomes showed no evidence of acute brain or behavioral effects or more enduring changes 1-month post-placebo; however, order confounds, such as practice or habituation effects, cannot be entirely discounted. It should be noted that the primary hypotheses of this paper were not pre-registered and *P* values were adjusted within all outcomes, but (due to the study's exploratory nature and design) not across outcomes. Between-subject confirmatory studies and superior imaging methodologies (e.g., multi-shell DTI) are now required to examine the reliability of the present study's findings.

## Methods

### Experimental model and subject details

**Ethical approvals and drug procurement.** This study was approved by the London-Surrey Research Ethics Committee and sponsored by the Joint Research and Compliance Office, Imperial College London. The National Institute for Health Research/Wellcome Trust Imperial Clinical Research Facility (ICRF) provided site-specific approvals. This research was carried out in accordance with Good Clinical Practice guidelines. All participants provided written informed consent.

An algorithm provided by the UK Medicines and Healthcare products Regulatory Agency (MHRA) was followed to assess whether the present study should be designated as a clinical or non-clinical trial. The algorithm and the MHRA confirmed that the present study was not a clinical trial. This was further confirmed by the sponsoring institution (Imperial College London) in the process of their sponsorship. Their decisions justify the designation of this study as an exploratory, translational study in healthy volunteers. In accordance with this designation, the study was not pre-registered as a clinical trial.

Bottled and encapsulated (size 4) psilocybin was provided by COMPASS Pathways and securely stored in Imperial College London in light-protected and temperature-controlled conditions. A Home Office Schedule 1 Drug License for the storage and handling of psilocybin was obtained. Psychotherapeutic and capsule (size and color matched) ingestion procedures were consistent across dosing days.

**Study design and participants.** This controlled, fixed-order, within-subjects study investigated the effects of psilocybin in healthy human adults with no prior psychedelic experience ($N = 28$). Participants were recruited using a convenience sampling approach through study advertisements and referrals. All participants received two oral doses of psilocybin, 4-weeks apart: (1) a control dose of 1 mg psilocybin on

the first dosing day, considered to be a subthreshold dose that is unable to occasion a psychedelic experience; and (2) a fully active dose of 25 mg psilocybin, considered to be a high dose and capable of inducing profound psychedelic effects, 4-weeks or 1-month later. This fixed order design was necessary given the hypothesized carry-over effects of 25 mg psilocybin. To uphold blinding and control for expectancy effects, participants were informed that they would receive psilocybin on both sessions of a variable dose up to 25 mg. No further information regarding dosage was provided. Participants were not financially compensated, but travel expenses were reimbursed. Data were collected between September 2016 and October 2020. For a schematic of the design and timeline of interventions, see Fig. S1 and Table S2.

Participants ($N = 28$) had an average age of 41 years (SD = 8.7, range: 29–59) and equally represented gender ($\chi^2 = 0.57$, $p = 0.450$) and educational attainment ($\chi^2 = 0.62$, $p = 0.430$). Sex and gender were recorded by self-report. Gender identity was concordant with sex assigned at birth for all participants. The study was not designed or powered to examine sex- or gender-based differences, and therefore no sex or gender analyses were conducted. All participants were naïve to psychedelic drugs and the majority were British (75%; $\chi^2 = 7.00$, $p < 0.01$) and Caucasian (86%; $\chi^2 = 14.57$, $p < 0.001$) in full-time employment (86%; $\chi^2 = 14.29$, $p < 0.001$). A breakdown of the participant demographic details can be found in Table S3. Recruitment information and full inclusion and exclusion criteria are detailed in the supplementary file.

### Methods Details

**Acute dosing procedures.** Participants refrained from caffeinated products and consumed a very light breakfast >1 h before arrival on dosing days. Upon arrival, participants were breathalyzed and provided a urine sample for drugs of abuse and pregnancy (where applicable). Participants set an intention for the psilocybin sessions before dosing, which was written on a white board in the dosing room. Dosing took place in a dimly lit, esthetically pleasing room within the ICRF facility. A music playlist was created on Spotify (https://open.spotify.com/playlist/7xcb0s46JhffYOY8euOEFO?si=1dc0dd49ac8a44dd) and simultaneously played through both headphones (Bose Sound-link II around ear) and speakers (KRK.RP5G3 Rockit Powered Studio). Participants wore a MindFold eye mask and were encouraged to lie in supine position on the bed. Both dosing days involved non-directive, guided supervision throughout. At the end of dosing days, participants were assessed for potential discharge by the study physician.

**EEG procedure.** To assess acute brain effects, a 24-channel wireless EEG cap (DSI-24 System, Wearable Sensing) with 21 dry electrodes (0.317 μV resolution, 300 Hz sampling rate and with an integrated amplifier, was used on dosing days. Electrodes were positioned following the 10–20 international format: Fp1, Fp2, Fz, F3, F4, F7, F8, Cz, C3, C4, T7/T3, T8/T4, Pz, P3, P4, P7/T5, P8/T6, O1, O2. Electrode Pz served as an online reference, with data re-referenced to electrodes A1 and A2 placed on the earlobes for the subtraction of noise in other channels. Fpz was selected as the ground electrode. All data were recorded using a Bluetooth-connected DSI-Streamer-v.1.08.41.

Four eyes-closed resting-state EEG (rsEEG) recordings, each lasting 4 min, were completed in silence at baseline (pre-dosing) and at -1 h, -2 h, and -4.5 h post-dosing. All EEG data were preprocessed using the Fieldtrip toolbox[66] in MATLAB (R2019B, MathWorks, Inc). rsEEG data were band-passed filtered at 1-45 Hz and visually inspected for gross artifacts. Noisy segments and channels were removed and interpolated, respectively. Independent Component Analysis (ICA) was used for the removal of eye blink artifacts. Next, data was re-referenced to the average and segmented in 2 s-epochs. Following preprocessing, spectral analysis was performed using Slepian multi-tapers with spectral smoothing of ±0.5 Hz using the Fieldtrip

toolbox[66]. Resulting spectra were divided into the following frequency bands for statistical analysis: *Delta* (1–4 Hz), *theta* (4–8 Hz), *alpha* (8–13 Hz), *beta* (13–30 Hz), and *gamma* (30-45 Hz). Scalp-level analysis of peak effects of 25 mg on EEG measures were performed using paired-*t*-tests at each electrode. Multiple comparisons of EEG results were controlled using cluster randomization analysis with an initial cluster-forming threshold of $p = 0.05$ repeated for 5000 permutations.

Please note, while an EEG task probing visual long-term potentiation was included[67,68], analysis revealed no significant potentiation of ERP components at a group level indicating unsuccessful implementation of the paradigm across conditions, and therefore no subsequent analysis was conducted.

**EEG LZc models**. To investigate if acute brain effects were associated with subsequent cognitive and behavioral changes, a procedure was designed to build a predictive model of a given outcome based on the values of a predictor at different electrodes and timepoints. In particular, our conjecture was that the dynamical trajectory of LZc could predict the level of psychological insight rated by participants via the PIS the day after dosing. The results shown in Fig. 3A and 3C are relevant in this regard. Figure 3D uses values across *all* sensors but at the 2 h timepoint when we know the intensity of subjective effects and plasma concentrations of psilocybin's active metabolite, psilocin, are maximal[69].

Data-driven analyses using acute rsEEG LZc for predicting psychological outcomes (insight and well-being changes) were carried out in a four-step procedure:

1) First, correlations between the target variable and LZc values across subjects were computed for each electrode and for each time point, separately. All electrode-timepoint combinations were ranked according to their correlation with the target variable.

2) Next, we constructed a predictor based on the average LZc of all electrode-timepoints whose correlation to the target variable was above a given value $\vartheta$. The performance of this predictor was assessed using out-of-sample $R^2$, computed via leave-one-out cross-validation. Specifically, we used $n-1$ subjects to (i) find the electrode-timepoint combinations that exceed the threshold, (ii) take the average LZc of each of the $n-1$ subjects in those electrodes, (iii) build a regression model of these values against the target variable, and then (iv) make a prediction for the target variable for the left-out subject. This procedure is repeated to obtain an out-of-sample residual (i.e., prediction error) for each subject, and the $R^2$ was calculated by comparing the variance of the residuals vs the variance of the target variable.

3) Then, we found the optimal threshold $\vartheta^*$ as the value of $\vartheta$ that maximized the out-of-sample $R^2$.

4) Finally, we built a final predictor following the same algorithm as in step 2, using all $n$ subjects and the optimal correlation threshold $\vartheta^*$. The performance of this predictor was assessed by calculating the correlation between predicted values and the target using Spearman's $r_s$, and its significance through surrogate data tests (i.e., comparing the out-of-sample $R^2$ of the model with the same quantity evaluated on data where the target values have been shuffled between subjects, repeated for 1000 permutations). These values are reported in the inset of Fig. 3A and 4C.

Please note that it is well known that the method used to choose the threshold—done in this case in step (3)—is known to not affect the false positive rate (i.e., the estimated statistical significance) of the test, but only its power (i.e., its sensitivity)[70].

To test the robustness of the findings obtained via this pipeline, we also investigated the results obtained via a simpler procedure: testing the predictive ability on both insight and well-being changes of simple spatial averages of the LZc values across all of the electrodes at different timepoints (i.e., 1 h, 2 h, and 4.5 h post-dosing). The findings of these additional analyses—with their significance corrected for multiple comparisons via Bonferroni—directly support the findings obtained with the data-driven method described here. These additional, simpler control analyses are reported in the supplementary material, Section 3.4.

**MRI/neural outcomes**. To assess enduring brain effects, participants attended three scanning sessions, 1-month apart: scan 1 collected baseline data, scan 2 served as the 1-month follow-up for 1 mg control as well as a baseline for post-25mg interventions, and scan 3 served as the one-month follow-up for 25 mg psilocybin and was the primary study endpoint. Imaging was performed on a 3 T Siemens Tim Trio using a 12-channel head coil. Whole-head anatomical images were acquired using the Alzheimer's Disease Neuroimaging Initiative, Grand Opportunity (ADNI-GO)[71] recommended Magnetization Prepared Rapid Gradient Echo (MPRAGE) parameters – 1 mm isotropic voxels, TR = 2300 ms, TE = 2.98 ms, 160 sagittal slices, 256 × 256 in-plane FOV, flip angle = 9°, bandwidth = 240 Hz/pixel, parallel imaging (PI) factor = 2, Inversion time = 900 ms. The three anatomical images from the different scanning data were merged into one T1 image, so to have better structural resolution. This was done by registering scan 2 and scan 3 to scan 1, and then averaging the scans together.

**Functional MRI procedures**. T2*-weighted echo-planar images (EPI) were acquired using the MB2R2 protocol[72] with interleaved slice acquisitions for the functional scans using 3 mm isotropic voxels, TR = 1250 ms, TE = 30 ms, 44 axial slices, 192 mm in-plane FOV, flip angle = 80°, bandwidth = 2232 Hz/pixel, GRAPPA acceleration = 2, number of volumes = 384. Both the resting state and emotional faces paradigm scans were each 8 min in duration. Resting-state scans were completed with eyes-closed.

**Pre-processing**. The preprocessing pipeline used here was similar to our previous studies[24,30], yet with slight modifications (e.g., not doing slice time correction due to the multiband sequence). FMRIB Software Library (FSL)[73], Analysis of Functional NeuroImages (AFNI)[74], Freesurfer[75] and Advanced Normalization Tools (ANTS)[76] were used to analyze the resting-state data. Motion was measured using frame-wise displacement (FD)[77]. The criterion for exclusion was subjects with >20% scrubbed volumes when the scrubbing threshold is FD = 0.4. Two subjects were excluded to high levels of head motion, and 23 subjects were used for the final analysis. The following preprocessing stages were performed: 1) removal of the first three volumes; 2) de-spiking (3dDespike, AFNI); 3) motion correction (3dvolreg, AFNI) by registering each volume to the volume most similar, in the least squares sense, to all others (in-house code); 4) brain extraction (BET, FSL); 5) rigid body registration to anatomical scans (FSL, BBR); 6) non-linear registration to 2 mm MNI brain (Symmetric Normalization (SyN), ANTS); 7) scrubbing[78] using an FD threshold of 0.4 (the mean percentage of volumes scrubbed for scan 1, scan 2, and scan 3 was $0.9 \pm 1.6\%$, $0.9 \pm 1.4\%$, and $2.6 \pm 5.3$, respectively). Scrubbed volumes were replaced with the mean of the surrounding volumes. Additional preprocessing steps included: 8) spatial smoothing (FWHM) of 6 mm (3dBlurInMask, AFNI); 9) band-pass filtering between 0.01 and 0.08 Hz (3dFourier, AFNI); 10) linear and quadratic de-trending (3dDetrend, AFNI); 11) regressing out 9 nuisance regressors (the same bandpass filter was applied on the nuisance regressors): out of these, 6 were motion-related (3 translations, 3 rotations) and 3 were anatomically-related (not smoothed). Specifically, the anatomical nuisance regressors were: 1) ventricles (Freesurfer, eroded in 2 mm space), 2) draining veins (DV) (FSL's CSF minus Freesurfer's Ventricles, eroded in 1 mm space) and 3) local white matter (WM) (FSL's WM minus Freesurfer's subcortical gray matter (GM) structures, eroded in 2 mm space). Regarding local WM regression[79], AFNI's 3dLocalstat was used to

calculate the mean local WM time-series for each voxel, using a 25 mm radius sphere centered on each voxel.

**Seed-based resting state functional connectivity (RSFC).** Based on prior hypotheses, four seeds were chosen for these analyses: (i) bilateral parahippocampus (PH), (ii) bilateral amygdala, (iii) ventromedial prefrontal cortex (vmPFC), and (iv) subgenual anterior cingulate cortex (sgACC). The PH seed was constructed by combining the anterior and posterior parahippocampal gyrus from the Harvard-Oxford probabilistic atlas, which was then thresholded at 50%. The vmPFC seed was the same as one previously used by our team in analyses of the acute effects of LSD, psilocybin and MDMA[24]. The sgACC seed was a 5 mm sphere centered at MNI coordinates ±2 28 −5. Bilateral amygdala seed was based on Harvard-Oxford probabilistic atlas, threshold at 50%. Mean time-series were derived for these seeds for each resting-state scan. The RSFC analyses were performed using FSL's FEAT for each subject. Pre-whitening (FILM) was applied. A higher-level analysis was performed to compare pre-treatment (scan 2) and post-treatment (scan 3) conditions using a mixed-effects GLM (FLAME 1 + 2), cluster corrected ($Z > 2.3$, p < 0.05). MRIcron was used to display the results. Only the amygdala results were significant.

**Emotional faces paradigm procedures.** The emotional faces paradigm was a block-design task lasting 8 min. Participants used a mirror mounted on the head-coil to view a screen mounted in the rear of the scanner bore, where visual stimuli were back-projected through a wave-guide in the rear wall of the scanner room. Participants were shown faces with either fearful, happy, or neutral expressions, selected from the Karolinska Directed Emotional Faces set[80]. An equal number of male and female faces were selected for the task. Each face was presented on screen for 3 s, and five faces of the same expression were presented in each 15 s block. Rest blocks (also 15 s) were also included, and there were 8 repetitions of each block type, presented in a pseudo-random sequence (32 blocks in total). Three versions of the task were used and the order of the task versions on each scanning visit was counter-balanced across participants. Participants passively viewed the faces but were instructed to press a single button with their thumb with the presentation of each new face, to confirm that they were paying attention to the stimuli.

For the emotional faces paradigm, we used the same resting-state preprocessing pipeline outlined above but with one modification—as in our previous emotional faces research[29]. The scrubbing threshold was increased to 0.9, as this is a threshold that better suits task paradigms[81]. No subjects were excluded due to head motion, and 25 subjects were used for the final analysis. The mean percentage of volumes scrubbed for scan 1, scan 2, and scan 3 was 1.1 ± 1.8%, 1.1 ± 2.3%, and 1.1 ± 1.9, respectively.

Different approaches were used to investigate changes in amygdala response in this paradigm: (i) voxelwise analysis within a bilateral amygdala mask (Harvard-Oxford atlas, probability >50%); (ii) calculating mean amygdala signal of left and right amygdala ROIs; and (iii) A whole-brain voxelwise analysis. For all approaches, a standard GLM was used for the first analysis step, as implemented in the FEAT module in FSL. Regressors derived from the onset times of each stimulus condition were convolved with a Gamma function in order to simulate the Haemodynamic Response Function (HRF). Prewhitening (FILM) was applied to correct for autocorrelations. Contrasts were defined that isolated activity related to each stimulus condition (fearful, happy, neutral) relative to the baseline, and comparisons were also made that contrasted between stimulus conditions, as appropriate i.e., [fearful > happy+neutral] and [happy > fearful+neutral]. Mixed-effects GLM (FLAME-1 + 2) was used for the voxelwise analysis with a statistical threshold of Z > 2.3, (cluster-corrected for multiple comparisons, $p < 0.050$).

**Generalized psychophysiological interaction.** To identify condition-specific modulation of amygdala functional connectivity (FC), we utilized a generalized psychophysiological (gPPI) approach using the CONN fMRI toolbox (https://www.nitrc.org/projects/ conn). Pre-processing steps included anatomical component correction (aCompCor) to remove noise and movement confounds on a voxel-to-voxel basis; denoising with parameters for white matter (5P), cerebrospinal fluid (5P), realignment (6P) and scrubbed volumes; bandpass filtering (0.008-0.09 Hz), and linear detrending after regression. The gPPI was computed using bivariate regression in CONN by specifying the following regressors: post-1mg fear, post-1mg happy, post-1mg neutral, post-25mg fear, post-25mg happy, post-25mg neutral, time-series regressor for the seed region (left and right amygdala), and PPI terms for the fear, happy, and neutral conditions. To minimize the number of statistical tests, only significant condition contrasts from the BOLD analysis were entered into analyses. Left and right amygdala seeds were run separately with alpha set at a false discovery rate (FDR)-corrected "$q$" value less than 0.05. FDR correction ($q_{FDR} = 0.05$) was also used to adjust for laterality (left and right amygdala seeds), and for the 3 condition contrasts of interest (fear, fear > happy + neutral, and happy > fear + neutral).

**Brain network modularity.** Brain network modularity was computed on cortical ($200 \times 200$)[82] interregional RSFC. As summarized by the Q value (see Star Methods), modularity is a measure of the decomposability of brain connectivity into distinct modules, where each module represents a set of brain regions that exhibit strong RSFC with each other (intra-modular RSFC) and weaker RSFC with regions of other modules (inter-modular RSFC).

For the modularity analyses, the RS-fMRI timeseries data were parcellated into 200 regions based on the Schaefer local-global parcellation[82] and Pearson's ($r$) correlation was computed between regional timeseries to create a $200 \times 200$ RSFC matrix. Modularity was then computed on this (unthresholded) matrix using the Louvain algorithm[83] implemented with the Network Community Toolbox (http://commdetect.weebly.com/), treating negative values asymmetrically. This algorithm finds modular partitions of the network (i.e., RSFC matrix) which optimize the modularity value, $Q$, by grouping nodes into non-overlapping modules (sets of regions) that maximize intra-modular and minimize inter-modular connections[84]. The modularity value $Q$ for a given modular partition is computed as follows:

$$Q = \frac{1}{l} \sum_{i,j \in N} \left[ w_{ij} - \frac{k_i k_j}{l} \right] \delta_{m_i m_j}$$

where $w$ is the edge weight (i.e., functional connectivity value) between nodes $i$ and $j$, $l$ is the sum of all weights in the graph, $k_i$ is the weighted degree (edge weight summed across all edges) of node $i$, and $m_i$ is a module containing node i. $\delta_{m_i m_j} = 1$ if nodes $i$ and $j$ belong to the same module, and = 0 otherwise. The $Q$ value for a given partition, therefore, quantifies the strength of within-module edges relative to the strength of between-module edges, or, in other words, the extent to which distinct modules can be delineated in the data. This algorithm has a single free parameter, gamma ($\gamma$), which controls how many modules will be detected. We kept this at the default value of 1. The Louvain algorithm was run iteratively 100 times at the individual-subject level and the average Q value across these runs was used for each subject. Given that modularity is sensitive to the average correlation strength of the network, $Q$ values were normalized for each subject by the mean $Q$ value generated by 100 randomly shuffled ("rewired") permutations of that subject's RSFC matrix.

**Structural MRI procedures.** Diffusion-weighted MRI (dMRI) data were acquired with the following acquisition parameters: 64 directions with b = 1000 s/mm²; 6 images without diffusion-weighting; 1 image without

diffusion-weighting and with the phase encoding direction reversed; TE = 88 ms; TR = 3010 ms; voxel size = 1.9 × 1.9 × 2.0 mm³; 72 slices. The acquisition was repeated three times to maximize signal-to-noise ratio (SNR). The images were preprocessed using MRtrix3[85] and the following steps: random matrix denoising[69,70], Gibbs-ringing reduction[86], distortion and motion correction[73,87], and bias field correction[88].

For distortion correction, MRtrix's "dwifslpreproc" was used to call FSL's "eddy" and "topup". "eddy" was used to correct eddy current-induced distortions and subject movements, while "topup" was used to correct susceptibility-induced distortions.

Region of interest (ROI) analysis was chosen for studying white matter plasticity. TractSeg's pre-trained neural network[89] was used to segment white matter and its outputs, representing probabilities of the tract being present in a voxel, were converted into binary masks with a threshold of 0.975 for reliable and reproducible segmentation. The following association fibers were included in the analysis: arcuate fascicle, cingulum, inferior occipito-frontal fascicle, inferior longitudinal fascicle, middle longitudinal fascicle, superior longitudinal fascicle, and uncinate fascicle. Furthermore, the following were included to incorporate the major frontal lobe tracts: anterior thalamic radiation, rostrum of the corpus callosum, genu of the corpus callosum, striato-prefrontal tract (PFC-STR), and thalamo-prefrontal tract (PFC-THA). ROIs from the different hemispheres were merged.

The diffusion tensor was estimated in each voxel using an iterated weighted least squares algorithm in Mrtrix3[85,90,91]. The mean values of axial diffusivity (AD) and radial diffusivity (RD) were calculated in each ROI for each subject and time point. Statistical analysis was performed on AD and RD that measure diffusion along orthogonal directions and are independent. In addition, diffusion tensors were estimated with a free-water component, done to remove partial volume effects[92]. Repeated measures ANOVA was performed on AD and RD separately for each tract with Bonferroni multiple comparisons correction. The analysis was then repeated for mean diffusivity (MD) and fractional anisotropy (FA).

A study-specific template was generated, which was used as the registration target in a Tract-Based Spatial Statistics (TBSS) pipeline[93]. A white matter skeleton was created from the averaged and registered FA images, which was thresholded at FA < 0.2. DWI metrics were then projected onto the skeleton to perform statistical analysis using FSL randomize.

## Psychological outcomes

**Cognitive flexibility.** Cognitive flexibility was measured via an intra-dimensional/extra-dimensional (IDED) task that had been optimized for internet-based delivery[36,37]. The IDED task consists of nine stages that require simple discrimination learning, compound discrimination learning, abstraction, attentional set-shifting, and reversal learning (see supplementary material for more details). Two stimuli are presented to the participant at a time. At any given phase, a hidden rule determines the correct response to the trial. Participants are required to determine the rule and choose the correct stimuli in six consecutive trials before progressing to the next phase. Participants are not informed when the rule changes, only receiving ticks or crosses as feedback for correct or incorrect responses, respectively. Participants fail the task upon responding incorrectly 30 times within any given phase. Stimuli sets are pseudorandomized (different sets of lines and shapes from a given pool) between timepoints. Errors made at each phase were adjusted to the total number of stimuli presented within each phase, representing a measure of "phase-accuracy" in order to assess changes in IDED performance across time-points while controlling for individual variability in learning rates. This adjustment was made by dividing the number of errors made by the number of trials completed per phase, therefore providing a representation of the percentage of trials in which an error was made. Participants completed the IDED task 1-day before and 1-month after each dosing day.

**Insight.** Psychological insight was measured via the Psychological Insight Scale (PIS)[34]. The PIS is a 6-7 item questionnaire that measures psychological insightfulness following a psychedelic experience (PIS-6) and accompanied behavioral changes (PIS-7). The PIS is scored using a VAS (0–100, with incremental units of one) with zero defined as 'no more than usual' and 100 defined as 'much more than usual'. Participants completed the PIS at 1-day, 2-weeks and one-month after each dosing day.

**Well-being.** Psychological well-being was measured using the 14-item *Well-being Warwick-Edinburgh Mental Wellbeing Scale* (WEMWBS)[35]. The WEMWBS is designed to assess mental well-being itself and not the determinants of mental well-being – e.g., resilience, problem solving, etc. The WEMWBS includes hedonic (i.e., happiness, life satisfaction) and eudaimonic (i.e., positive relationships, psychological functioning) items, which together measure mental well-being. Items are rated on a 5-point Likert scale (from 1 = none of the time to 5 = all of the time) to yield a total summed score, with a minimum possible score of 14 and a maximum score of 70 (population norms: $M = 51$, SD = 9)[94]. Participants completed the WEMWBS 1-day before and at 2-weeks and 4-weeks after each dosing day.

**Retrospective ranking of the experience.** Broadly inspired by (but not directly related to) previous methods by Griffiths et al.[95], 6 months post-dosing, participants reflected on the peak of each dosing session and answered the following three questions: 1) "How profound was the state of consciousness you experienced?", 2) "How intense was the state of consciousness you experienced?", and 3) "How unusual was the state of consciousness you experienced?" (See Fig. S20). Each item was rated using an eight-point Likert scale, from 1 = "no more 'profound/intense/unusual' than routine everyday states of consciousness" to 8 = "the single most 'profound/intense/unusual' state of consciousness of my life, that I can recall").

## Quantification and statistical analysis

All statistical analyses were performed using either SPSS version 26.0 (IBM Corp., Armonk, NY, USA), R Studio 2022.07.1 (RStudio PBC, Boston,. MA, USA), SciPy 1.9.1 and statsmodels 0.13.2 in Python 3.9.13 (Python Software Foundation), or MATLAB 2019b (MathWorks, Natick, USA) and illustrated using GraphPad Prism version 9.2 (GraphPad Software, La Jolla California USA). Data were subjected to one- and two-way repeated measures ANOVAs or linear mixed-effects (LME) model, where appropriate, and followed up on with *post-hoc* tests corrected for multiple comparisons within all outcomes. The structure of random effects in LME models was built following standard recommendations[96]. To determine sphericity, the Greenhouse-Geisser procedure was used to estimate epsilon (i.e., Greenhouse-Geisser Estimate Epsilon ($\varepsilon$)) and used to correct the degrees of freedom of the F-distribution when $\varepsilon < 0.75$. Where the assumption of sphericity was violated, the $\varepsilon$ values and corrected p-values are reported. The 95% CIs are provided. Effect sizes were calculated using the repeated measures Cohen's *d* formula. Statistical significance was assumed at the p ≤ 0.050 probability level. The non-parametric cluster test of the correlation matrix in Fig. 3B was done following the procedure described in[70,97], using a critical threshold of α = 0.05.

## Reporting summary

Further information on research design is available in the Nature Portfolio Reporting Summary linked to this article.

# Data availability

The self-report, behavioral, processed EEG, and processed MRI data supporting all analyses in this manuscript are included in the Source Data file. Due to their large size and the need for detailed guidance

regarding preprocessing and analysis, access to the raw and high-dimensional imaging datasets is restricted. These data are available from the authors upon request. Access is limited to academic, non-commercial research purposes. Requests should be directed to Taylor Lyons (taylorlyons@me.com), Richard J. Zeifman (rzeifman@newschool.edu), or Robin L. Carhart-Harris (robin.carhart-harris@ucsf.edu), and should include a brief description of the proposed use. The authors aim to respond to requests within one week. Approved users will be granted access for a period of 24 months. Source data are provided with this paper.

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

## Acknowledgements

This work was funded by philanthropic donations to the Centre for Psychedelic Research, Imperial College London. T.L. received an MRC-DTP Studentship, H.M.D. received an Imperial College London President's PhD Scholarships. R.C.H. was supported by the Alex Mosley Charitable Trust and the Ralph Metzner Distinguished Professorship (UCSF). Further support was provided by The Beckley Foundation. Infrastructure support was provided by the NIHR Imperial Biomedical Research Centre and the NIHR Imperial Clinical Research Facility. The views expressed are those of the author(s) and not necessarily those of the NHS, the NIHR or the Department of Health and Social Care.

We thank Manca Peskar, Victoria Nygart and Mellissa Shukuroglou for their invaluable assistance on study visits; Dr. Roberta Murphy and Dr. Jonny Martell for clinical oversight; Dr. Rosalind Watts for advice, training and support on guiding and integration procedures; Dr. Robin Tyacke for controlled drug oversight; Joe Peill, Gregory Cooper, Katie Trinci and Pablo Mallaroni for their support with data analysis; and Bruna Giribaldi for advice and support.

## Author contributions

R.C.-H. designed the study. R.C.-H. and D.J.N. provided oversight. T.L. coordinated the study, led the study visits, and collected the data. M.S. coordinated the study to completion, leading study visits and collecting data. L.E., L.R., C.T. and F.R. supervised dosing sessions and performed data collection. L.E. and D.E. provided medical cover and assisted with study visits. T.L. analyzed the behavioral data and conducted regressions and correlations between the metrics. F.R. and P.M.M. ran the clustering analyses of long-term changes and completed the path model. L.K. and L.R. analyzed the diffusion MR and functional MR data, respectively. F.R., P.M.M., F.S., C.T., M.S., and H.D. analyzed the EEG data. R.C.H. and T.L. wrote the manuscript. All authors provided edits and accepted the final draft.

## Competing interests

The authors declare the following competing interests: R.C.-H. is a scientific advisor to Atai Beckley, Otsuka, and Entropy Neurodynamics. D.J.N. is a scientific advisor to COMPASS Pathways, Neural Therapeutics, Algernon Pharmaceuticals and Alvarius. R.J.Z. has received financial compensation for serving as a consultant to Beckley Clinical. R.C.-H., D.J.N., and R.J.Z. declare no additional competing interests. L.T., M.S., L.K., F.E.R., L.R., P.A.M.M., C.T., L.O., B.A.P., A.H., W.T., H.M.D., M.G., K.G., H.K., F.S., L.E., A.G., M.B.W., and D.E. declare no competing interests.

## Additional information

[1]Division of Psychiatry, Department of Brain Sciences, Centre for Psychedelic Research, Imperial College London, London, UK. [2]UCL Great Ormond Street Institute of Child Health, University College London, London, UK. [3]Department of Informatics, University of Sussex, Brighton, UK. [4]Data Science Institute, Imperial College London, London, UK. [5]Centre for Complexity Science, Imperial College London, London, UK. [6]Centre for Eudaimonia and Human Flourishing, University of Oxford, Oxford, UK. [7]Department of Psychology, University of Cambridge, Cambridge, UK. [8]Department of Psychology, Queen Mary University of London, London, UK. [9]School of Psychology and Australian Institute for Bioengineering and Nanotechnology (AIBN), University of Queensland, Brisbane, QLD, Australia. [10]NYU Langone Center for Psychedelic Medicine, Department of Psychiatry, NYU Grossman School of Medicine, New York, NY, USA. [11]Department of Psychology, The New School for Social Research, New York, NY, USA. [12]Department of Neuroimaging, Institute of Psychiatry, Psychology & Neuroscience, King's College London, London, UK. [13]Department of Brain Sciences, Imperial College London, London, UK. [14]Department of Neurology and Neurosurgery, Montreal Neurological Institute, McGill University, Montreal, QC, Canada. [15]Carhart-Harris Lab, Department of Neurology, University of California San Francisco, San Francisco, CA, USA. [16]Neuroscape, Department of Neurology, University of California San Francisco, San Francisco, CA, USA. [17]Invicro London, Hammersmith Hospital, Faculty of Medicine, Imperial College London, London, UK. ✉e-mail: robin.carhart-harris@ucsf.edu

