## [Transparent Peer Review file · Nature Communications]

Human brain changes after first psilocybin use

Corresponding Author: Dr Richard Zeifman

Version 0:

Reviewer comments:

Reviewer #1

(Remarks to the Author)

This study examines anatomical and functional changes in the brain from one-hour to one-month following a 25mg psilocybin administration. The authors also report changes in cognitive flexibility, psychological insight, and well-being. This is an intriguing dataset probing clinically relevant research questions, yet I have concerns about the associated analyses and interpretations that I believe the authors need to consider.

The Introduction is very brief. The authors state they are addressing important knowledge gaps, but do not make clear what those are, leaving little context for the analyses reported. I suppose the brevity is due to manuscript length constraints, yet I think the manuscript suffers from the lack of a concrete introduction to guide the reader.

On line 69 the authors write no changes in EEG were observed following 1mg psilocybin. The analyses reported here and in Table S5 and Figure S6 revolve around dose-x-time interaction effects, which are relative changes, so it is not clear what reported estimate supports that there is no change following only 1mg psilocybin. Please clarify how this statement is supported and provide relevant details in the manuscript. This sentiment is again alluded to in the Discussion, where again it is articulated without support.

Please provide some indication of whether p-values are adjusted or unadjusted. There are allusions to doing so in the Methods and sometimes something is articulated in the Results but inconsistently, making it unclear for which any adjustment is applied.

The authors describe evaluating effects on PFC-STR and PFC-THA AD tracts. Based on Figure S18, these two tracts appear to overlap quite a bit. This makes me wonder how distinct they are, that is, to what extent they represent PFC-STR or PFC-THA AD, specifically, or whether PFC-STR and PFC-THA AD each represent essentially the same construct of AD along this path, which encompasses many tracts. Can the authors elaborate on how similar these two AD estimates are and what implications that has for their interpretation as two separate outcomes?

Within the "Brain response to emotional faces" and "Resting-state FC (RSFC)" sections of the results the authors appear to report statistics determined from mean estimates from clusters identified by related voxel-wise analyses. Do I understand this correctly? If the statistical test for defining the clusters is related to the statistical analysis applied to the extracted values, then the effect sizes are upwardly biased and the significance estimates invalid. This is even more perplexing because, for example, the legend for Figures S8 and S14 seems to acknowledge this confound; the authors write in those legends that they refrain from statistical analyses, yet they are reported in the main text. Please clarify the source of the reported estimates and how the reported statistical estimates relate to comments in the supplemental figure legends that state why estimates are not reported.

Figure S16 is described in the main text (line 124) as a significant interaction, yet it is not clear, what term is moderating the effect of Timepoint? It is not clear how this test across timepoints highlights the "robustness" of the one-month 1mg and one-month 25mg comparison earlier in this section of the Results. The former is an ANOVA across three timepoints whereas the latter is a comparison of two of those three time points, in what way does this establish "robustness"?

There are errors in how supplementary figures are referenced. For example, the legend for Figure S9 alludes to elements of

Figure S7A and S7B that do not seem to match what is represented in Figure S7. Within Supplementary 4.3 the authors allude to Figure S7A where I think they mean Figure S10. The dizzying number of supplementary figures is not benefited by inaccurate references between them. Please ensure the figures are correctly referenced.

Please provide some description of the non-significant effects of modularity, for example, effect sizes. This is helpful for prospective future studies.

Please provide some rationale for why to apply FDR correction in some contexts and FWER (e.g., Bonferroni) in others? It reads confusingly to switch between the two without a clear rationale. Further, they appear to combine them on line 228, which would seem to create a significance estimate with a particularly complicated interpretation. These unexplained variations in the statistical approaches make the paper read disjointedly.

Related, the authors mention a particular correction method, Greenhouse-Geiser, based on an undescribed epsilon term indicating an undescribed violation of sphericity (line 225). Please describe this consideration in the Methods.

The change in cognitive performance measures evaluated is seemingly problematically confounded by the fixed-order study design. The authors report a significant decrease in EDS errors (line 230) following the 25mg dose. I do not see how the current approach can distinguish whether this is due to drug exposure or getting better at the task with practice. The authors pay little attention to the cognitive performance findings in the Discussion, and include only a passing half phrase about order effects on line 415. I do not see how the authors can present the cognitive performance results without acknowledging or addressing this confound.

Please describe in the Methods “non-parametric cluster statistics” described on line 246. Regardless of the correction, a small p-value does not “imply a non-spurious correlation structure,” I suggest the authors rephrase this statement.

The data driven analyses described beginning on line 253 do not seem “data driven”. On line 533, the authors indicate the model is both trained and applied “using all n subjects.” This is the same as a typical correlation estimate across a dataset and if data driven at all, it is overfit to the current data. For example, if this was the method used to identify the three electrodes described in Figure 4A, this would seem an overfitting of these data. I suggest the authors consider an alternative phrasing to “data driven” or a method that does not train and test on the same data.

The citation associated with the paragraph beginning on line 541 does not clearly relate to the content of the paragraph.

Considering the author’s apprehension about the interpretation of DTI effects (line 322), could they suggest prospective future studies that could corroborate the current observations? Related, it reads oddly for the authors first to express “serious caution” in interpreting the DTI (line 340) then proceed to interpret them in the next sentence.

Please clarify where the lack of relation between EEG complexity and DTI changes is reported; this is mentioned on line 353, but I do not see the related results.

What is meant by “atypicality” on line 389? Do the authors mean symptom severity or some specific symptom dimension?

Minor considerations:

Why is neuroplasticity in quotations (line 32)?

The authors use LZ and LZc for Lempel-Ziv complexity. I suggest using only one to avoid confusion.

There are multiple places where abbreviations are re-defined or used prior to definition, please scan the article to remove these instances.

I think the incongruence between the order of results and order of supplementary materials undermines the readability of the manuscript. I suggest reordering the supplementary materials to align with the order in which results are presented.

Line 229, it seems the authors mean “Figure S22”, not “Figure 22”.

What is meant by the asterisk in Figure S25?

Reviewer #2

(Remarks to the Author)

This manuscript reports on a very interesting trial with first time psilocybin users. The report appears to contain the full spectrum of measures that were taken. It should however be stated explicitly if additional measures were taken or if this is all.

While I like that the authors do not follow a publication strategy where 10th of papers are made out of one dataset, I believe the short format of the given manuscript suffers a bit from the super short introduction, where the reasoning for choosing the one or other measure (explanation of the operationalization) is not well explained.

Overall, I hope to see this study reported in a high impact journal. However, I have two main concerns:

- I did not find any preregistration of the analyses and the test for the specific hypotheses. As there is hundreds of test

possible, I would consider it highly important to have the main correlations between different measures preregistered to make sure there is no suspicion on cherry-picking possible. I feel most tests can well be motivated, however, we know that one can always find some post-hoc motivations if one is searching for them. Therefore a preregistration would substantially strengthen the given implications.

- Correspondingly it should be made more explicit how the authors deal with multiple comparison correction (See further comments)
- I believe the choice of a 8min emotional face paradigm was not a good one. I think this data should be dropped from the report. One does not really learn much from it and the data is noisy.

In the following I have collected comments about the manuscript, which the authors might want to address.

Abstract

- The authors should not just report that there was an effect on “decreased axial diffusion”, but explain what this is the operationalization for, and why this is relevant and what it means.

Introduction

- I like that it is short. However it does not explain why the different measures have been chosen and applied and what operationalization was applied where. E.g. the operationalization of “neuroplasticity”, or why do you actually measure the specific functional fMRI task

Results

- Could the authors clarify if LZ might be prone to influences of artefacts from measurement noise, and if so, how? Does it play a role here that maybe in the drug session there was more motion, potentially? So how can you exclude that it is lower data quality during the drug session that drives the effect?
 - I do not understand the comparison for multiple comparison, nor do I understand the hypotheses for the DTI Analysis
 - Reporting that a whole brain analysis yielded a “large effect” in functional task-based fMRI is not a suited analysis. The data appears noisy – this part of the study is poorly motivated and there is very little data – I would recommend to drop this part from the report
 - RSFC are at the same time reported to be explorative (?exploratory?) and hypothesis driven – be more clear here and make clear if the hypothesis driven analyses have been pre-registered, otherwise I do not see that it is reasonable to report them as such. For a seed-based analysis you would do a whole brain Family wise error correction – it is unclear what the authors mean by doing a Bonferroni correction here.
 - Network Modularity: The “introductory information” is provided in the methods section instead having it in the introduction. It does not become clear if the “controlling for wellbeing”-analysis was preregistered or is well motivated.
 - Figure 2, C is not a typical way of displaying such results and is pretty noisy.
 - The correlation in Figure 2 C, is not really convincing, even though significant. But not convincing for the scope of the interpretation.
 - Figure3: Mark the significances in the figure and not on the x-axis...
 - With regards to psychological wellbeing: The figure display cuts off the axis without explicit display of the range of possible ratings – you should improve this display by labelling minimum and maximum values the y-axis value could take
 - For the Cognitive Flexibility test the authors suddenly switch to apply FDR correction instead of Bonferroni correction – why? Usually one does not mix different correction levels – it contributes a bit to the perception of the article, that multiple people did different analysis which were copy&pasted together without proper cross-adjustments
- #### Discussion
- Overall I like the neutral position in the discussion which in a balanced way presents previous reports without overselling and overinterpretation of them
- The authors state that there were no long-term changes after 1mg. I feel the corresponding analyses are not clear to me, in order to make the statement so strong for a null effect.
 - It would have been nice to see if there is a correlation between the subjective experiences and the acute brain entropy changes, as this is considered a key mechanism. Did you also assess the phenomenology in a bit more detail and could report it? Or was it only the Intensity rating, which is very little and then a very limiting factor of the study
 - Could you report more of control measures to assure the data quality in EEG and fMRI Measures is equal across conditions? There might be training, habituation, differences in locomotion due to the experience, which might all critically influence the quality of these signals. It would be good to see quality measure comparisons like the amount of noise quantified and motion parameter summaries...

Methods

- I understand that the study was not preregistered a clinical trial – but was it preregistered at all?
- I don't understand what balanced with regards to education attainment means, and what the test statistic represents, as there was only one group?
- I do not understand the exclusion of participants from the fMRI analysis. You say overall there were 28 and excluded 2 and used 23 for the analysis?
- With regards to different numbers of subjects in different analyses, you could try to be more clear in the report of results
- When reading that the emotional face paradigm was only one run of 8min, I am not surprised about the noisiness of the data. I have a strong opinion that one should not collect and not use such data (even it exists in the literature with different patient groups). It is noisy and unreliable. Such short functional runs simply cannot provide reliable data. In order to get convinced here, I feel multiple proof-of-principle analysis would be necessary, like showing baseline contrasts and a very good alignment with the literature. As mentioned above, I would discard this data from the report and just decide this data is not worth reporting
- The authors report to also increase the scrubbing threshold here, which indicates there was more movement. Actually, for task-based fMRI it is very unusual to do scrubbing (this is more typical for RSFC Analysis).

- Could you elaborate how the measure of LZ might be specifically sensitive to changes in the Alpha range or not?

Statistical Treatment and preregistration of analyses

- Already in the Abstract it becomes clear that many measures were taken, which gives rise to an exploratory search for predictors of treatment outcome, on the neural and the behavioral level. From the write up of the article at hand, it does not become clear to me which analyses were preregistered in which way, where the reports of the non-significant analyses are and if the type of error correction for multiple comparisons were done accurately. If there is no sufficient preregistration and correction, the results are still very interesting, but need to be framed much more clearly as exploratory to make sure that readers are aware of potential alpha inflation when interpreting the report

- Whenever you do a "Bonferroni Correction" please specify for how many tests you corrected. There is large variability in what people consider a "family of tests", so I believe the strength of being convincing of your finding will benefit from making this explicit.

Supplement

Requires more careful compilation of materials – figures are not optimized, naming of figures are confused, some supplementary material is not clear what it is supposed to do. Appears a bit rushed and almost careless, which is not strengthening my confidence in the overall carefulness of the given analyses.

- With regards to supplementary table S5 (EEG Outcomes), the authors might think about some more efficient way of display – it takes a bit of time to get orientation in this table.
- Figure S6 would benefit from error margins on the frequency plot and a labelling of the power spectrum (in the display) to specify from which electrodes this spectrum is derived. Also the color of the lines does not match the color in the legend
- Figure S7 should be generated again – it looks like it has been stretched in y-direction
- Figure S8: Is section A indeed reported at $p < 0.05$ uncorrected? The resulting regions appear unreliable and then simply extracting z-scores from them not a suited analysis. If the main effects are not significant – what does this tell? I do not understand the overall purpose of this figure, except showing non-significant results. (In general for extraction of effect sizes it might be useful to use RFX Plot instead of reporting z-scores.
- Figure S9 should be generated again – it looks stretched in the x-direction; also please adjust figures according to size so that they are matched... (simply for aesthetic reasons – this causes me pain to look at) – also here – rather extract effect sizes with RFX plot. Indicating the significant comparison would be classic double dipping – so also delete the bracket. Do not say "response to fear" which is conceptually not meaningful.
- In section 4.3, the authors refer to figure S7A which does not exist. The effects reported in 4.3 are not valid. You need to test for the difference in the reduction between 1mg and 25mg.
- 4.4 The threshold of 50% is very conservative and not plausible to me. Also the formulations is strangely complicated: "we threshold at 50% probability and then extract the values from the entire left amygdala, given that threshold" – overall the initial responses are weak and noisy, none of the contrasts is showing a relevant effect. This should be noted. You might want to combine S10 and S11 to make the visual inspection and comparison for consistency more straight forward
- 4.6 There is inconsistent information, when you say that it is the effect of high-dose psilocybin and then specify that it is the contrast "post-1mg > post-25mg. Tables should be tables and not screenshots which are hard to read and inconsistent in the display and formatting. What does the "q-" in "q-FDR" stand for? The results of the gPPI should be displayed on a whole brain without masking, to provide the reader with some feeling of how noisy these analyses are and if the given results are potentially a selective report.
- Also the format and the font sizes and the general style of Figure S13 should be adjusted to make it more readable. I know this is a supplement – but still you should not drop figures here, where one cannot read the axis labels, and cannot see anything in the renderings on the brain. The figure caption is not readable and a different style than the others. Overall appearing the authors copy&paste from different source and do not put the effort to connect and synchronize data treatment and presentation. In this figure it is completely unclear to me from what analysis the different figures are derived and what is what... So please redo the entire figure to make the presentation of results more clear. This includes a clearer naming of the seed regions, a whole brain rendering of all results and then a clear indication for which subregions you are plotting effect sizes (these are most likely circular analyses)
- Figure S14: adjust your figures in design and do not make blue lines on top of black lines. Unclear what the figure actually shows. "cluster-corrected map" is unclear in its meaning – also is it again really at $p < 0.05$ uncorrected? Then it is not significant and no result – simply noise.
- All analyses here seem to be without appropriate correction for multiple comparison and therefore not worth reporting
- 4.11: I would believe that finding different anxiety levels invalidated all analyses and results which include comparisons to baseline
- Figure S18 should be made more effective by introducing headlines and introducing better what is the difference between the four columns
- S23. Figure dimensions distorted... introduce abbreviation in legend to make it somehow meaningful

Reviewer #3

(Remarks to the Author)

Please see the attached document for the review.

Reviewer #4

(Remarks to the Author)

I co-reviewed this manuscript with one of the reviewers who provided the listed reports. This is part of the Nature

Communications initiative to facilitate training in peer review and to provide appropriate recognition for Early Career Researchers who co-review manuscripts.

Version 1:

Reviewer comments:

Reviewer #1

(Remarks to the Author)

The authors have adequately addressed my comments.

Reviewer #2

(Remarks to the Author)

Overall the responses to reviewer comments are not always constructive.

If one is suggesting to include some information in the abstract, it is not appropriate that the authors simply answer, that there is a word limit and they will not do it.

The exploratory nature of the study is now clearly labelled within the text – however, this should also be mentioned in the Abstract already

Overall, I believe that both reviewers agreed that one is getting lost in the amount of analyses and how the authors are dealing with multiple comparisons.

I overall come to the conclusion that this manuscript is overloaded for the given format. It is not possible to tie together all of these test and analyses and it does not get better with review rounds.

So for me it is overloaded and ultimately it is not understandable how all of these results belong together. A so long supplement with more and more analyses does not help to understand which are simply “additional nice to have analyses” and which of the analyses support the main claim of the article.

I think it is ok to do this – but for me it is no clear enough for a format with such high visibility, considering that the overall study was exploratory in its nature and not preregistered.

Other points:

I do not understand the response of the authors to Reviewer #1 requests about double-dipping. This requires more clarification why their procedures would not be double-dipping

The responses to reviewer#1 request if an ANOVA is appropriate to test for robustness, is surprising – the authors might want to suggest alternative analyses

The supplement is still having figures with bizarre bad resolution (e.g. S9), which require correction – I believe supplements are typically submitted as PDF – so this is the job of the authors

Line 351: „of the psychedelic trip“ should be reformulated to something like “induced subjective experienced altered states effects...” or some other formulation which points to the fact that it is about subjective experience, and avoiding the unscientific term “trip”.

Related: I would encourage the authors to discuss more that there was only the measure of “intensity of drug effects” assessed. Which is a shame. Please verify that indeed there were no other measures taken for different types of subjective experiences. You might refer to other studies which do this with much more detail and explain why you did not do this, and discuss this limitation. Mainly: That one cannot say which aspect of the experience was relevant, as “intensity” is not really informative.

Reviewer #3

(Remarks to the Author)

The authors have satisfactorily addressed this reviewer's comments and concerns. Looking forward to the publication of this interesting contribution to the literature.

Reviewer #4

(Remarks to the Author)

REVIEWER COMMENTS

Reviewer #1 (Remarks to the Author):

This study examines anatomical and functional changes in the brain from one-hour to one-month following a 25mg psilocybin administration. The authors also report changes in cognitive flexibility, psychological insight, and well-being. This is an intriguing dataset probing clinically relevant research questions, yet I have concerns about the associated analyses and interpretations that I believe the authors need to consider.

Thank you for reviewing this manuscript and for your fair and well considered comments and suggestions.

The Introduction is very brief. The authors state they are addressing important knowledge gaps, but do not make clear what those are, leaving little context for the analyses reported. I suppose the brevity is due to manuscript length constraints, yet I think the manuscript suffers from the lack of a concrete introduction to guide the reader.

We have now expanded the introduction. We say more about knowledge gaps, e.g., that we are unclear regarding long-term brain changes, and give some context to the outcomes we examine.

On line 69 the authors write no changes in EEG were observed following 1mg psilocybin. The analyses reported here and in Table S5 and Figure S6 revolve around dose-x-time interaction effects, which are relative changes, so it is not clear what reported estimate supports that there is no change following only 1mg psilocybin. Please clarify how this statement is supported and provide relevant details in the manuscript. This sentiment is again alluded to in the Discussion, where again it is articulated without support.

Table S5 contains both the marginal effects of time in the low-dose condition as well as the interaction effect of high-dose x time (see the first three columns of Table S5). We appreciate this point can be made clearer, so we have relabeled the column headers of this table, and we now refer to these explicitly in the relevant segments of the manuscript.

Please provide some indication of whether p-values are adjusted or unadjusted. There are allusions to doing so in the Methods and sometimes something is articulated in the Results but inconsistently, making it unclear for which any adjustment is applied.

P values are adjusted within all outcomes but not across outcomes. We now make this clearer in the relevant sections of the paper (see line 393 of the methods). We are also transparent about this in the limitations section of the discussion (see lines 462-464), as we are about the lack of pre-registered primary hypotheses.

The authors describe evaluating effects on PFC-STR and PFC-THA AD tracts. Based on Figure S18, these two tracts appear to overlap quite a bit. This makes me wonder how distinct they are, that is, to what extent they represent PFC-STR or PFC-THA AD, specifically, or whether PFC-STR and PFC-THA AD each represent essentially the same construct of AD along this path, which encompasses many tracts. Can the authors elaborate on how similar these two AD estimates are and what implications that has for their interpretation as two separate outcomes?

Yes, we make the same observation. In the supplement, we provide images showing the tracts together and separately, so that the reader can discern the extent of the convergence versus divergence. We now explicitly make the point in the text (Line 127-130, Page 5) that the tracts are largely converging/overlapping and can be interpreted as such.

Within the "Brain response to emotional faces" and "Resting-state FC (RSFC)" sections of the results the authors appear to report statistics determined from mean estimates from clusters identified by related voxel-wise analyses.

There was cluster correction using standard thresholds ($Z=2.3$, $p<0.05$).

Do I understand this correctly? If the statistical test for defining the clusters is related to the statistical analysis applied to the extracted values, then the effect sizes are upwardly biased and the significance estimates invalid.

Yes, this is correct. The so-called "double-dipping" phenomenon, widely recognized to be methodologically problematic. We have been very careful not to fall foul of this statistically offence.

This is even more perplexing because, for example, the legend for Figures S8 and S14 seems to acknowledge this confound;

That's correct.

the authors write in those legends that they refrain from statistical analyses, yet they are reported in the main text.

This was an error. We have now removed this and instead merely report the functional MRI tests as yielding largely negative and statistically weak effects.

Please clarify the source of the reported estimates and how the reported statistical estimates relate to comments in the supplemental figure legends that state why estimates are not reported.

The reason is due to a prior review process at another journal. There, a reviewer rightly critiqued an analytic approach that could have been accused of “double dipping”. The reviewer recommended we not report statistical tests. We agreed with this suggestion and complied. We now merely report that effects failed to survive appropriately statistical thresholds for significance. We show the plots in figure S8 because they are sufficiently informative to include, albeit with the relevant caveats and no statistical tests shown or inferences drawn. Usefully, the plots show the effect in the fear condition relative to the other conditions. The plots could be regarded as having illustrative value.

Figure S16 is described in the main text (line 124) as a significant interaction, yet it is not clear, what term is moderating the effect of Timepoint?

Timepoint subsumes the effect of drug (25mg) versus either baseline or 1mg (placebo). The data shown in S15 shows that the most robust contrast is 1mg versus 25mg, which is what you’d infer if the 25mg psilocybin was truly driving the effect.

It is not clear how this test across timepoints highlights the “robustness” of the one-month 1mg and one-month 25mg comparison earlier in this section of the Results.

Perhaps the term “robustness” isn’t optimal. We had used it here because the relevant effect is significant in both the interaction analysis and the posthoc pairwise contrasts (S14-16). Another way to state it is that 1mg vs 25mg is the contrast with the largest effect.

The former is an ANOVA across three timepoints whereas the latter is a comparison of two of those three time points, in what way does this establish “robustness”?

We are open to an alternative term. Accordingly, we’ve now replaced “robust” with “significant”.

There are errors in how supplementary figures are referenced. For example, the legend for Figure S9 alludes to elements of Figure S7A and S7B that do not seem to match what is represented in Figure S7.

Sorry, these files underwent a series of revisions. We have now corrected these errors.

Within Supplementary 4.3 the authors allude to Figure S7A where I think they mean Figure S10. The dizzying number of supplementary figures is not benefited by inaccurate references between them. Please ensure the figures are correctly referenced.

Apologies again. We can confirm that the Figures and Tables are now correctly labelled and referenced.

Please provide some description of the non-significant effects of modularity, for example, effect sizes. This is helpful for prospective future studies.

Now done. Please note, however, that when reviewing the stats for this analysis, it became apparent that the baseline vs post 25mg contrast was actually significant. The plot and text has now been revised accordingly. The relevant text can be found on line 181-187.

Please provide some rationale for why to apply FDR correction in some contexts and FWER (e.g., Bonferroni) in others? It reads confusingly to switch between the two without a clear rationale. Further, they appear to combine them on line 228, which would seem to create a significance estimate with a particularly complicated interpretation. These unexplained variations in the statistical approaches make the paper read disjointedly.

Given the large number of comparisons involved in the analysis of performance across multiple stages of the IDED, we applied FDR correction. This approach was chosen to balance false positives and power, to mitigate the risk of Type II errors (false negatives) from Bonferroni correction in the context of the IDEDs numerous statistical tests (i.e. 27 comparisons). FDR is the standard correction approach commonly used for the IDED in the literature (e.g. Luo et al., 2022; Langley et al., 2020; Tyagi et al., 2019; Hampshire et al., 2010 etc.), hence our use of it here. Elsewhere, we used Bonferroni as a rule. The only exception was for the PPI analysis, for which Bonferroni has now been applied for hemispheric correction.

Related, the authors mention a particular correction method, Greenhouse-Geiser, based on an undescribed epsilon term indicating an undescribed violation of sphericity (line 225). Please describe this consideration in the Methods.

We have now included relevant text in the methods section. All new text is highlighted in green.

The change in cognitive performance measures evaluated is seemingly problematically confounded by the fixed-order study design. The authors report a significant decrease in EDS errors (line 230) following the 25mg dose. I do not see how the current approach can distinguish whether this is due to drug exposure or getting better at the task with practice. The authors pay little attention to the cognitive performance findings in the Discussion, and include only a passing half phrase about order effects on line 415. I do not see how the authors can present the cognitive performance results without acknowledging or addressing this confound.

Order confounds were recognized as a limitation in the original discussion (now highlighted). However, we now add a second component to this sentence “e.g., practice or habituation effects”. However, it should be noted that while the IDEED can be susceptible to practice effects in the early stages of learning and during intradimensional shifts, the extradimensional shift stage requires a shift to a completely novel dimension and is generally less affected by prior practice, retaining test retest reliability (see Lowe & Rabbitt (1998) –Test/re-test reliability of the CANTAB and ISPOCD neuropsychological batteries: theoretical and practical issues. *Neuropsychologia* 36(9): 915-923) – it therefore remains a reliable indicator of cognitive flexibility. Further, the change after post 1mg versus post 25mg is the largest ($d = 0.6$). This arguably supports an effect of 25mg versus a mere practice effect, as practice should also be apparent in baseline versus post-1mg (yet that contrast was not significant). This point is now made in the relevant figure legend. All said, however, we still highlight the single-arm, fixed order study design as a potential general limitation in the paper’s conclusions.

Please describe in the Methods “non-parametric cluster statistics” described on line 246. Regardless of the correction, a small p-value does not “imply a non-spurious correlation structure,” I suggest the authors rephrase this statement.

Thank you for the nuanced point. We are confident that our test is correctly assessing statistical significance i.e., that we can infer that the result is not due to chance, but we have revised to text to say this more precisely. As a principle of frequentist statistics, the size of the p value (0.006) is irrelevant. To make the process more transparent we have added a sentence in the “Quantification and statistical analysis” section of the Methods.

The data driven analyses described beginning on line 253 do not seem “data driven”. On line 533, the authors indicate the model is both trained and applied “using all n subjects.” This is the same as a typical correlation estimate across a dataset and if data driven at all, it is overfit to the current data. For example, if this was the method used to identify the three electrodes described in Figure 4A, this would seem an overfitting of these data. I suggest the authors consider an alternative phrasing to “data driven” or a method that does not train and test on the same data.

We have now rephrased this section, removing the reference to the method being “data-driven”.

The citation associated with the paragraph beginning on line 541 does not clearly relate to the content of the paragraph.

It is not clear what paragraph is being referred to here. Please clarify.

Considering the author’s apprehension about the interpretation of DTI effects (line 322), could they suggest prospective future studies that could corroborate the current observations?

Yes, we now do this. But please note that we do say this in the discussion section: “Further research using multi-shell sequences is needed to disambiguate the current findings and inform on their robustness and replicability.”

Further, we thank the reviewer for highlighting this. We agree that future studies are essential to corroborate and extend our current findings. In addition to the point already made in the Discussion (highlighting the need for multi-shell diffusion imaging) we have now added further prospective directions. These include the application of advanced models of tissue microstructure (e.g., NODDI), the inclusion of longitudinal follow-up scans beyond one month, and integration with behavioral and pharmacokinetic data to better link structural changes to therapeutic mechanisms. This has been added to the revised Discussion section as follows:

“Specifically, incorporating multi-shell diffusion imaging protocols would enable more biologically specific modelling of white matter changes (e.g., via neurite orientation dispersion and density imaging, or NODDI), as well as longitudinal follow-up beyond one month to examine the persistence or progression of microstructural effects. Integrating these with detailed behavioral outcomes and pharmacokinetic measurements may also clarify the functional relevance of anatomical changes and the neurobiological mechanisms underlying therapeutic response.”

Related, it reads oddly for the authors first to express “serious caution” in interpreting the DTI (line 340) then proceed to interpret them in the next sentence.

There's a catch 22 at play here. We feel we are right to advise caution while also offering some interpretation. Previously, we refrained from making any interpretation at all but were asked to by a previous reviewer to offer some interpretation. A solution here is to advise the caution and have it as a caveat when considering the interpretation. Further, we have now revised the text in the discussion to emphasize the tentative nature of these interpretations and to ensure that they are presented as such.

Please clarify where the lack of relation between EEG complexity and DTI changes is reported; this is mentioned on line 353, but I do not see the related results.

We report the result as negative but don't provide values for the test.

What is meant by “atypicality” on line 389? Do the authors mean symptom severity or some specific symptom dimension?

We now clarify this: “by which we mean that baseline values are farther from a healthy population average”.

Minor considerations:

Why is neuroplasticity in quotations (line 32)?

“Neuroplasticity” was written in inverted commas as it is arguably over-used jargon until precisely defined. We now address this and remove the quotations and instead are more specific as to what we're referring to when we use the term. Specifically, we say: “increased properties of anatomical neuroplasticity, such as synaptogenesis.”

The authors use LZ and LZc for Lempel-Ziv complexity. I suggest using only one to avoid confusion.

Thank you. Now corrected. We used LZc consistently in the main paper but (appropriately) not the supplement.

There are multiple places where abbreviations are re-defined or used prior to definition, please scan the article to remove these instances.

Ok, thanks. Now corrected, we believe.

I think the incongruence between the order of results and order of supplementary materials undermines the readability of the manuscript. I suggest reordering the supplementary materials to align with the order in which results are presented.

Line 229, it seems the authors mean “Figure S22”, not “Figure 22”.

Correct. Now corrected.

What is meant by the asterisk in Figure S25?

This asterisk was redundant and has now been removed.

Thank you for your thorough review. You have enabled us to improve our work.

Reviewer #2 (Remarks to the Author):

This manuscript reports on a very interesting trial with first time psilocybin users. The report appears to contain the full spectrum of measures that were taken. It should however be stated explicitly if additional measures were taken or if this is all.

While I like that the authors do not follow a publication strategy where 10th of papers are made out of one dataset, I believe the short format of the given manuscript suffers a bit from the super short introduction, where the reasoning for choosing the one or other measure (explanation of the operationalization) is not well explained.

Overall, I hope to see this study reported in a high impact journal.

Thank you for this positive review and appreciation of our comprehensive reporting of outcomes.

However, I have two main concerns:

- I did not find any preregistration of the analyses and the test for the specific hypotheses. As there is hundreds of test possible, I would consider it highly important to have the main correlations between different measures preregistered to make sure there is no suspicion on cherry-picking possible. I feel most tests can well be motivated, however, we know that one can always find some post-hoc motivations if one is searching for them. Therefore a preregistration would substantially strengthen the given implications.

You are correct that this is a legitimate limitation. You are also correct that the non-CT status of the work and that it was principally exploratory does not excuse the absence of pre-registered primary hypotheses. We now make this clearer in the discussion section.

- Correspondingly it should be made more explicit how the authors deal with multiple comparison correction (See further comments)

- I believe the choice of a 8min emotional face paradigm was not a good one. I think this data should be dropped from the report. One does not really learn much from it and the data is noisy.

We now recognize this limitation in the discussion. Line 520-521. We have elected not to drop the paradigm entirely from the report but instead, in the spirit of transparency, we describe it in full so that readers can digest and critique the methodology and results.

In the following I have collected comments about the manuscript, which the authors might want to address.

Abstract

- The authors should not just report that there was an effect on “decreased axial diffusion”, but explain what this is the operationalization for, and why this is relevant and what it means.

This is true but word count prevents us from unpack such things within the abstract.

Introduction

- I like that it is short. However it does not explain why the different measures have been chosen and applied and what operationalization was applied where. E.g. the operationalization of “neuroplasticity”, or why do you actually measure the specific functional fMRI task.

You're correct. We've now been explicit regarding what we mean by “neuroplasticity” and we also offer more of a justification for our outcomes more generally.

Results

- Could the authors clarify if LZ might be prone to influences of artefacts from measurement noise, and if so, how?

LZc is generally only affected by noise in the direction of decreasing effect sizes i.e., as most sources of noise introduce statistical regularity, they diminish the effect of drug in promoting irregularity or entropy.

Does it play a role here that maybe in the drug session there was more motion, potentially?

As above, motion shouldn't translate into greater LZc values but rather, lower ones.

So how can you exclude that it is lower data quality during the drug session that drives the effect?

Because when we clean noisy epochs and regress sources of noise, ensuring data quality is maintained across both conditions. Furthermore, typical EEG artefacts (such as heartbeats, eye movements, blinks, etc) are either periodic or slow, and therefore they would lower entropy (LZc), rather than increase it.

- I do not understand the comparison for multiple comparison, nor do I understand the hypotheses for the DTI

Analysis

- Reporting that a whole brain analysis yielded a “large effect” in functional task-based fMRI is not a suited analysis.

A comment like this would only ever be made in relation to a large effect size e.g., a large Cohen's d value. Indeed, it is being used in that context here. Thus, it's not being said in relation to an analysis per se but the size of an effect and more specifically, the (by convention) large effect size of $d = 0.9$.

The data appears noisy – this part of the study is poorly motivated and there is very little data – I would recommend to drop this part from the report

It's unclear what you're suggesting we drop. It's also unclear what the basis is for the claim of "noisy" data.

- RSFC are at the same time reported to be explorative (?exploratory?)

Correct, "exploratory" is better. Now changed.

and hypothesis driven – be more clear here and make clear if the hypothesis driven analyses have been pre-registered, otherwise I do not see that it is reasonable to report them as such.

No, they weren't pre-registered.

For a seed-based analysis you would do a whole brain Family wise error correction – it is unclear what the authors mean by doing a Bonferroni correction here.

For the seed-based analyses we did indeed use FWE correction. Specifically, we used a standard cluster-based correction ($Z=2.3$, $p < 0.05$) for the whole-brain maps. Then we used a Bonferroni correction for the individual statistical contrasts across the three time-points.

- Network Modularity: The "introductory information" is provided in the methods section instead having it in the introduction. It does not become clear if the "controlling for wellbeing"-analysis was preregistered or is well motivated.

Controlling for well-being is well-motivated. It's convention to control for baseline when examining change in mental health. We do this in trials. If not done, highly atypical baseline values bias change as per regression to the mean.

- Figure 2, C is not a typical way of displaying such results and is pretty noisy.

We have now revised Figure 2 accordingly.

- The correlation in Figure 2 C, is not really convincing, even though significant. But not convincing for the scope of the interpretation.

- Figure3: Mark the significances in the figure and not on the x-axis...

Figure 2 and 3 have been revised accordingly.

- With regards to psychological wellbeing: The figure display cuts off the axis without explicit display of the range of possible ratings – you should improve this display by labelling minimum and maximum values the y-axis value could take

Fair point. We now state the range in the figure legend.

- For the Cognitive Flexibility test the authors suddenly switch to apply FDR correction instead of Bonferroni correction – why? Usually one does not mix different correction levels – it contributes a bit to the perception of the article, that multiple people did different analysis which were copy&pasted together without proper cross-adjustments

This is fair. We did work as a large team, with specialists working on specific components e.g., Adam Hampshire is the cognitive scientist that advised on the cognitive flexibility paradigm and analyzed it as per standard practice. Given the large number of comparisons involved in the analysis of performance across multiple stages of the IDED, we applied FDR correction. This approach was chosen to balance false positives and power, to mitigate the risk of Type II errors (false negatives) from Bonferroni correction in the context of the IDEDs numerous statistical tests (i.e. 27 comparisons). FDR is the standard correction approach commonly used for the IDED in the literature (e.g. Luo et al., 2022; Langley et al., 2020; Tyagi et al., 2019; Hampshire et al., 2010 etc.), hence our use of it here. Elsewhere, we used Bonferroni as a rule. It's important to note that there are multiple sets of independent data in this paper. There is no reason to do cross-adjustments when data are statistically independent. They represent different 'families' (as in 'family-wise error correction') of tests, therefore no cross-adjustments are needed.

Discussion

Overall I like the neutral position in the discussion which in a balanced way presents previous reports without overselling and overinterpretation of them

Thank you for recognizing that.

- The authors state that there were no long-term changes after 1mg. I feel the corresponding analyses are not clear to me, in order to make the statement so strong for a null effect.

We can confirm that there was no evidence of long-term changes with 1mg psilocybin. The statement is simply accurate to the observation.

- It would have been nice to see if there is a correlation between the subjective experiences and the acute brain entropy changes, as this is considered a key mechanism. Did you also assess the phenomenology in a bit more detail and could report it? Or was it only the Intensity rating, which is very little and then a very limiting factor of the study

This is a very good suggestion. In addition to intensity, we began by looking at Insight as we had a prior hypothesis there. In fact, internally, the study was always known as the 'Insight study'. We plan to further investigate the acute brain correlates of subjective experience in a future study.

- Could you report more of control measures to assure the data quality in EEG and fMRI Measures is equal across conditions? There might be training, habituation, differences in locomotion due to the experience, which might all critically influence the quality of these signals. It would be good to see quality measure comparisons like the amount of noise quantified and motion parameter summaries...

For DTI quality control, we monitored motion parameters across all sessions. Motion correction was performed using FSL's eddy tool as described in the methods. Further, inspection of EEG and fMRI data confirmed no systematic differences in data quality between scanning sessions.

Methods

- I understand that the study was not preregistered a clinical trial – but was it preregistered at all?

No.

- I don't understand what balanced with regards to educational attainment means, and what the test statistic represents, as there was only one group?

Apologies, the wording used here was indeed confusing and has now been amended. The test statistic represents a calculation of within-sample differences in categorical variables. We found no differences between the number of males vs females or participants with university-level vs secondary school-level education within the sample. However, the sample overrepresents those that are Caucasian (86%), British (75%), and in full-time employment (86%).

- I do not understand the exclusion of participants from the fMRI analysis. You say overall there were 28 and excluded 2 and used 23 for the analysis?

28 subjects participated in the dosing sessions but n=3 could not complete their scanning sessions due to the Covid lockdown, leaving us with N=25 for MRI data. We have included this in a table within the supplement (see S27), which now states exact p numbers per analyses and reasons (if any) for exclusions.

- With regards to different numbers of subjects in different analyses, you could try to be clearer in the report of results

This is a fair request. As above, we now provide a table of the sample size for relevant metrics in the supplement (see S27).

- When reading that the emotional face paradigm was only one run of 8min, I am not surprised about the noisiness of the data.

The brevity of the task at 8 minutes does not naturally imply noise. Very generally, one can observe good effect sizes and highly reliable fMRI activation patterns with relatively short visual or motor tasks.

I have a strong opinion that one should not collect and not use such data (even it exists in the literature with different patient groups). It is noisy and unreliable.

This is an opinion to which you are of course entitled, however, it is one we do not share. Emotional face paradigms of this duration have yielded significant findings in our previous studies. We could always have chosen to provide more power but the lack of effect also implies a weak effect. It is therefore a null result worth reporting rather than "burying", albeit we now only show relevant maps and results in the supplement. Readers can decide how to interpret and we refrain from over-interpretation. We also now note in the discussion section that the paradigm may have been insensitive.

Such short functional runs simply cannot provide reliable data.

We would note that emotional face tasks are a very standard method in the literature, and many papers use tasks of fairly short duration. In particular we would mention two specific papers which have explicitly focused on test-retest reliability with this task. The first (<https://www.sciencedirect.com/science/article/pii/S1053811912001565>) used a task duration of 4m28s (as well as two other cognitive tasks) and noted “(1) all three tasks robustly activated their particular target regions; (2) the group-level activation maps were highly stable across sessions for all three tasks”. The second (<https://www.sciencedirect.com/science/article/pii/S1053811917304160>) used three face-related tasks which varied between ~4-6mins. While both these studies noted that individually-based reliability measures (ICC values) were generally somewhat poor, group-level activation maps were highly robust. This is very often the case with any fMRI paradigm. Our task compares favourably with these previous studies in terms of task duration, and we have also recently reported robust results from a near-identical task in a clinical sample (<https://psychiatryonline.org/doi/10.1176/appi.ajp.20230751>).

In order to get convinced here, I feel multiple proof-of-principle analysis would be necessary, like showing baseline contrasts and a very good alignment with the literature. As mentioned above, I would discard this data from the report and just decide this data is not worth reporting

We will comply with your request and move these data from the main manuscript.

- The authors report to also increase the scrubbing threshold here, which indicates there was more movement. Actually, for task-based fMRI it is very unusual to do scrubbing (this is more typical for RSFC Analysis).

We wanted to preserve a standard pipeline for all the fMRI data, so used motion-scrubbing for both task and resting scans. However, the reviewer is correct that motion-scrubbing is less commonly used for task data, so we took a less aggressive approach and raised the threshold to 0.9 (as recommended by <https://onlinelibrary.wiley.com/doi/epdf/10.1002/hbm.22307>, which we cite in the methods).

- Could you elaborate how the measure of LZ might be specifically sensitive to changes in the Alpha range or not? Yes, alpha is the most prominent brain-based statistical regularity in such datasets. Thus, if it drops— as it does under psychedelics— then it would follow that statistical irregularity (what LZc computes) would go up. The two aren't 1:1 100% inversely related but there is a natural relationship, nevertheless.

Statistical Treatment and preregistration of analyses

- Already in the Abstract it becomes clear that many measures were taken, which gives rise to an exploratory search for predictors of treatment outcome, on the neural and the behavioral level.

Yes, but all measures done are reported, including, as you can see, largely negative outcomes (fMRI).

From the write up of the article at hand, it does not become clear to me which analyses were preregistered in which way, where the reports of the non-significant analyses are and if the type of error correction for multiple comparisons were done accurately. If there is no sufficient preregistration and correction, the results are still very interesting, but need to be framed much more clearly as exploratory to make sure that readers are aware of potential alpha inflation when interpreting the report

This is fair. We're now clearer about the exploratory, hypothesis-generating nature of the study.

- Whenever you do a “Bonferroni Correction” please specify for how many tests you corrected. There is large variability in what people consider a “family of tests”, so I believe the strength of being convincing of your finding will benefit from making this explicit.

This is fair. We've now done this.

Thank you for your thorough and fair review.

Supplement

Requires more careful compilation of materials – figures are not optimized, naming of figures are confused, some supplementary material is not clear what it is supposed to do. Appears a bit rushed and almost careless, which is not strengthening my confidence in the overall carefulness of the given analyses.

As you can see, the supplement is an extensive document. It is the inclusiveness of the document that has led to the errors. However, we've now checked through it and believe we've corrected the errors.

- With regards to supplementary table S5 (EEG Outcomes), the authors might think about some more efficient way of display – it takes a bit of time to get orientation in this table.

Noted. It is a complex table. We have now changed the layout of the table while retaining the information.

- Figure S6 would benefit from error margins on the frequency plot and a labelling of the power spectrum (in the display) to specify from which electrodes this spectrum is derived. Also the color of the lines does not match the color in the legend

The figure legend has been amended to help clarify some of these issues. It's pasted below:

Figure S6. Maximum effects of 25mg vs 1 mg of psilocybin in EEG features (LZc and power in frequency bands) occurred 2-hours after administration. Localized scalp changes are marked with red electrodes are shown in the left ($p < 0.05$, two-sided, cluster-corrected). Red dots indicate electrodes where there was a significant difference between the conditions. Averaged power spectra of brain activity at 2h, is shown on the right for both 25mg (red) and 1mg (blue) conditions. The spectra on the right derive from an average of sensors.

- Figure S7 should be generated again – it looks like it has been stretched in y-direction

The information is accurate and the request is merely one of aesthetics, however, we have complied and corrected it.

- Figure S8: Is section A indeed reported at $p < 0.05$ uncorrected? The resulting regions appear unreliable and then simply extracting z-scores from them not a suited analysis. If the main effects are not significant – what does this tell? I do not understand the overall purpose of this figure, except showing non-significant results. (In general for extraction of effect sizes it might be useful to use RFX Plot instead of reporting z-scores.

It is important to show null findings. This is why it's being shown. Note also it's only being shown in the supplement. We extract from that contrast because it is the most salient but we are always transparent about the process. We favor this inclusive approach to reporting. If we did not report, we'd be open to the charge of burying negative results.

- Figure S9 should be generated again – it looks stretched in the x-direction; also please adjust figures according to size so that they are matched... (simply for aesthetic reasons – this causes me pain to look at) – also here – rather extract effect sizes with RFX plot. Indicating the significant comparison would be classic double dipping – so also delete the bracket. Do not say “response to fear” which is conceptually not meaningful.

Again, this is an aesthetic request. We feel the request is overkill for a supplement. The information is accurate. We are not making a formal comparison, so there can be no charge of “double dipping”. We're merely showing data. The bracket is there to be consistent with S8. The focus on that contrast is done as it is the most salient based on prior work, yet no stats are run to claim statistical significance.

- In section 4.3, the authors refer to figure S7A which does not exist. The effects reported in 4.3 are not valid. You need to test for the difference in the reduction between 1mg and 25mg.

Sorry, this was an error. It's now correct to S8A.

- 4.4 The threshold of 50% is very conservative and not plausible to me. Also the formulations is strangely complicated: “we threshold at 50% probability and then extract the values from the entire left amygdala, given that threshold” – overall the initial responses are weak and noisy, none of the contrasts is showing a relevant effect. This should be noted. You might want to combine S10 and S11 to make the visual inspection and comparison for consistency more straight forward

There's no formal basis for these claims of “noisy” data. There is always natural variance and group data like these. We make no claims or interpretations on the supplementary data other than to recognize weak and mostly null functional brain changes (fMRI). We continue to favor our inclusive policy on data reporting. The 50% threshold refers to the definition of the amygdala in the Harvard/Oxford probabilistic atlas, in this case we're selection voxels which have a >50% probability of being in the amygdala, which his conservative, and a standard approach.

- 4.6 There is inconsistent information, when you say that it is the effect of high-dose psilocybin and then specify that it is the contrast “post-1mg > post-25mg. Tables should be tables and not screenshots which are hard to read and inconsistent in the display and formatting. What does the “q-” in “q-FDR” stand for? The results of the gPPI should be displayed on a whole brain without masking, to provide the reader with some feeling of how noisy these analyses are and if the given results are potentially a selective report.

Thank you for catching this – we have adjusted the specified contrast in S13 to “post-1mg vs. post-25mg” to include both increases and decreases in connectivity. For clarity, we have also replaced “q-FDR” with “p-FDR” and indicated clusters that survived Bonferroni correction for laterality (right/left hemisphere) with an asterisk. This is the relevant text: “* = p-FDR < 0.025 (FDR corrected on brain level and Bonferroni corrected for laterality (left and right amygdala

seeds)". The gPPI results display the target clusters detected with the specified significance threshold. All clusters have been exhaustively displayed.

- Also the format and the font sizes and the general style of Figure S13 should be adjusted to make it more readable. I know this is a supplement – but still you should not drop figures here, where one cannot read the axis labels, and cannot see anything in the renderings on the brain. The figure caption is not readable and a different style than the others. Overall appearing the authors copy&paste from different source and do not put the effort to connect and synchronize data treatment and presentation. In this figure it is completely unclear to me from what analysis the different figures are derived and what is what... So please redo the entire figure to make the presentation of results more clear. This includes a clearer naming of the seed regions, a whole brain rendering of all results and then a clear indication for which subregions you are plotting effect sizes (these are most likely circular analyses)

Requests for better readability are fair. We've sought to do this with an improved S13. Please find that in our resubmission. I would, however, challenge the comment regarding not putting effort to synchronize data. The team effort has been immense on the project. Please consider that the point you are making is on a very extensive supplementary file, 38 pages of supplementary data.

- Figure S14: adjust your figures in design and do not make blue lines on top of black lines. Unclear what the figure actually shows. "cluster-corrected map" is unclear in its meaning – also is it again really at $p < 0.05$ uncorrected? Then it is not significant and no result – simply noise.

The blue line is not on black but on a gray brain image. It's a standard output from the imaging analysis software. The map is properly cluster corrected as stated in the text: "cluster-corrected map (threshold $Z = 2.3$, $p < 0.05$)". It is a significant result using standard thresholding therefore your point is invalid. To claim "noise" (again) is unjustified.

- All analyses here seem to be without appropriate correction for multiple comparison and therefore not worth reporting

This is incorrect. We used standard cluster correction as stated.

- 4.11: I would believe that finding different anxiety levels invalidated all analyses and results which include comparisons to baseline

That's a strong interpretation and it would not be ours. We feel justified in being so thorough and transparent in our reporting. It's good practice. And we do not agree with your interpretation.

- Figure S18 should be made more effective by introducing headlines and introducing better what is the difference between the four columns

The image has a title and each column has a sub-title.

- S23. Figure dimensions distorted... introduce abbreviation in legend to make it somehow meaningful.

We included the abbreviations in the original legend. Pasted below as evidence:

Figure S23. A) The IDED with its various sub-parameters shown. Abbreviations: Intra-dimensional/extra-dimensional task; SD, simple discrimination; C_D, compound discrimination with second dimension; CD, compound discrimination; IDS, intra-dimensional shift; EDS, extra-dimensional shift: SR, simple reversal; CR, compound reversal, IDR, intra-dimensional reversal; EDR, extra-dimensional reversal. EDS is considered the most relevant index of cognitive flexibility; this is the parameter we measure and refer to as "cognitive flexibility errors" in B & C. ** = $p < 0.01$. B) EDS with all data included, including an outlier participant ($M_{diff}=0.13$, $SE=0.04$, $p=0.008$; 95% CI [0.02, 0.12], $d=0.6$). ** = $p < 0.01$. C) We also conducted an outlier analysis using the ROUT method (for 'Robust regression and Outlier removal') which identified one significant outlier across two timepoints (i.e., pre-dose baseline and one-month post-1mg). Figure C therefore shows the results with this outlier removed. As this outlier scored in the direction of our findings, we report C in the main manuscript ($M_{diff}=0.06$, $SE=0.02$, $p=0.016$; 95% CI [0.00, 0.12], $d=0.5$), i.e., we report the more conservative analysis in the main paper and show here how the main result was not driven by the outlier. * = $p < 0.05$, ** = $p < 0.01$.

Thank you for your thorough review. It has helped improve the manuscript. We challenged some specific points regarding the supplement but only where they either missed an already included detail or were otherwise unjustified.

Reviewer #3 (Remarks to the Author):

Please see the attached document for the review. See 'PDF review' below.

Reviewer #4 (Remarks to the Author):

I co-reviewed this manuscript with one of the reviewers who provided the listed reports. This is part of the Nature

PDF review

Manuscript Review: Human brain changes after first psilocybin use

The authors present a multimodal analysis of the pre- and post- treatment effects of a single dose of psilocybin in psychedelically-naïve human subjects (N=28) on measures of EEG, fMRI and DTI using a placebo-controlled design and a repeated measures analysis. The 25mg dose induced a significant decrease in axial diffusivity in prefrontal-subcortical tracts that corresponded with decreased modularity when assessing the pre- and post- effects. Initial increased cortical signal entropy response at 1-2 hours predicted an overall increase in psychological well-being one month later.

Major Strengths and Innovation

- The authors build on previous identification of unique effects that psilocybin, a 5-HT_{2A}R agonist, has on the overall modularity and entropy of the whole brain to predict how acute brain entropy can predict psychological benefit up to a month post dosing.
- The authors use a functionally inactive placebo (1mg dose of psilocybin) as well as a crossover design to allow for validation of psychological, structural and functional brain changes within each participant while controlling for the issue of finding a psychedelic placebo.
- The authors use a combination of multimodal imaging techniques which is essential to obtain a well-rounded perspective of the varying degrees of change that occur in the brain after psilocybin ingestion, especially with regards to region specific changes, and acute vs longer lasting effects.
- This study uses a multi-modal imaging approach to better quantify the enduring effect of psilocybin after a single dose. This study not only demonstrates that acute entropy can predict specific psychological outcomes, but that psilocybin has an enduring effect on DTI measures of white matter diffusivity and anisotropy and white matter tractography with region specificity.

Thank you for this positive review and acknowledgement of our work.

Major Weaknesses

- The authors present a varied assortment of analyses and methods that do not always follow a logical order, especially in the context of how they relate to the overall hypothesis. A significant discussion and union of the analyses and their unified interpretation is also absent. Much of the data analysis seems to be exploratory rather than chosen with respect to an initial hypothesis. The paper seems exploratory to the

point of running superfluous statistical analyses to find significant relationships. The paper clearly references previous studies and similar work in the field, yet the main findings are relatively adjacent to these previous findings and do not significantly advance the state-of-the-art as a reader might expect to see in this journal.

The study was indeed exploratory. We would challenge the comment regarding there being no scientific advance here. If the white matter changes hold in new studies, then this will prove to have been a major discovery here. Further, in its own right, the identification of a predictive biomarker (LZc on EEG) of mental-health change could prove to be an exceptional breakthrough for the field of psychedelic medicine.

- This work is original and complements current minor gaps in the field of psychedelics in the context of neuroplasticity.

Please define “neuroplasticity”. Some would argue that without definition this term is over-used jargon.

However, the many different forms of statistical analyses paired with an unclear initial and end goal reduces the significance of this work and creates a constellation of findings that does not allow a significant impact.

We would challenge this conclusion.

- The conclusion and claims are supported by the various methods and analyses undertaken in the paper, although additional information relative to methodology may be needed to clarify why or how the underlying hypothesis was chosen. References are made to previous studies, but there is not enough discussion of how the new results affect the current field, which gaps are being challenged, and what the next steps in determining a new hypothesis could be.

We hope that our new revised introduction now provides a better context for the work that was performed here.

- It may be worthwhile to provide additional information about how exploratory this work was meant to be. There are places throughout the manuscript where it is unclear how the data analysis technique is chosen (Lines 371-373, 401-402). It may be useful to follow a similar format per section (fMRI vs DTI vs psychological assessment) to clearly re-state the data analysis steps and then conclude on the main finding. Overall, the interpretation of results is unclear and disordered, which points to a need to review both the introduction and discussion sections of the paper. The overall concluding statement relates to statistical choices rather than an overarching theme, or final major finding(s). This lack of structure and clear scientific method reduces the impact of the results.

The point is well received. We have now reframed the study and its aims to be clearer that it was exploratory and hypothesis generating. Please see our revised introduction.

Regarding the discussion, we elected to be cautious and not over-interpret. We've now gone back and revised the discussion to better highlight the salient findings and how they should motivate future trials.

- The methodology does not seem to follow a direction with a clear hypothesis and points to the study as being primarily explorative, especially considering the many different types of analyses performed. The paper has interesting perspectives, with the findings relative to fMRI, DTI and LZc and how it relates to psychological outcomes at one-month post-drug administration, yet they are minor and do not advance the field as much as they could.

This interpretation is fair and accurate. Please see how this is properly recognized now in our revised introduction.

- While the methodology and general analysis may not be as clear, the Methods are well detailed and contain enough detail that this work could be reproduced.

Thank you.

Statistical Analysis

- The authors utilized various stringent statistical analyses across data types, including paired t-test for EEG data, a combination of using residuals and correlation to demonstrate the relationship between LZc and psychological outcomes, a mixed-effects GLM to compare the effect of treatment in rs-fMRI and emotional paradigm scan sessions, a bivariate regression to consider the interaction of psychophysiological effects, permutations to analyze the rs-fMRI matrix, and a repeated-measures ANOVA with TBSS to study the change between treatment groups.

Thank you for recognizing this.

Line-specific corrections:

General comments:

Throughout the paper, there are references that appear as a placeholder for a complete sentence. It would be useful to insert references at the end of a full sentence to improve the flow of the writing.

Introduction

Line 28: Please double check the formula for psilocybin.

We believe the formulae were accurate but have now been written using a more formal nomenclature. Please see revised introduction.

Line 32: Throughout the manuscript, the references follow an incomplete sentence. Please consider reviewing proper citation methods and formatting.

Manuscript revised to try to address cases.

Line 36: The phrasing of this sentence implies that the LSD study involved depressed patients. Please consider rewriting this.

Now revised.

Lines 38-39: Citation format

Citations checked.

Line 41: Please define the knowledge gaps and how this relates to the hypothesis of the research study.

Now addressed. Please see highlighted text in revised introduction.

Line 45: Please clarify if this is rs-EEG or not.

It is. Now clarified.

Results

Line 65: Please consider explaining LZc in more detail here.

Now done. Please see revised Results section.

Line 82: Which ANOVA is used? This changes from ANOVA to inclusive ANOVA to repeated measures ANOVA to one-way ANOVA and is unclear.

It's all those things. The "inclusive" is perhaps redundant, it just means we used all of the data, as per the text "all factors, all levels."

Line 93: What feedback has been received? Is this based on result outcomes?

Sorry, it's unclear what this is referring to. I'm afraid the line number isn't matching our document and "feedback" couldn't be found in a text search.

Line 122: Inclusive ANOVA- does this refer to a repeated measures ANOVA?

Yes. No amended throughout to say that.

Line 155: Please consider explaining a bit more what modularity means.

Now done. Please see revised Results section.

Line 158: Please consider explicitly mentioning the figure within the Supplemental file.

S17. Ok. Now done.

Lines 189-190: Please consider explaining what IDED and EDS means, potentially introducing it before it appears in the figure legend.

This has now been addressed by simply moving the relevant results text above Figure 3.

Line 267: How was this normalized?

Values were zero-centered.

Lines 271-272: This is not a complete sentence.

Now completed. Please see revised discussion section.

Discussion

Line 301: Please consider reintroducing the overall experimental design and/or hypothesis driven question.

Summarized now.

Lines 303-305: It is unclear if this sentence is necessary to this paragraph.

Apologies, it's hard to follow what the specific text is that is being referred to. Line numbers have shifted.

Lines 313-316: Please consider rephrasing this, the meaning is not as clear as it could be.

Line 321: Please consider explaining why DTI measures of AD are chosen based on the previous literature describing spine formation in the frontal cortex/ synaptic density changes and why AD is chosen over other DTI measures. It may be judicious to avoid overinterpretation of AD as it is a minor DTI metric and the biological interpretation is unclear to this day.

This is fair. We do stress caution in interpreting the finding.

Line 338: This is the first mention of FA that seems to appear, why is this measure not discussed in more detail.

It is the second mention. We report it here in the results section: "and fractional anisotropy (FA) changes also became statistically significant (Figure S21)." FA only became significant after free-water correction. This correction is important as it removes confounding effects from extracellular free water, revealing tissue-specific changes. The convergent findings across both AD and FA after correction strengthen our interpretation of genuine microstructural alterations

Lines 339-340: Please consider rephrasing “under-myelinated axons”. This does not seem to be the correct terminology.

Thank you. We agree that the original phrasing was imprecise. We have revised the wording to more accurately reflect what is measurable with DTI, avoiding implied histological interpretation:

“involving newly forming, minimally myelinated axons, both of which could influence diffusion properties without necessarily reflecting mature white matter organization.”

Line 362: This is not a proper citation.

Lines 375-378: Please consider rephrasing this sentence.

Line 383: This is not a complete sentence.

It is difficult to follow these points due to the line number shift while editing.

Line 391: This section follows a section discussing fMRI. This could be moved to the beginning of the discussion, otherwise as is, this reads like the beginning of another paper. Please consider changing this to flow better.

We have now moved mention of fMRI to earlier in the discussion.

Lines 401-402: Please consider rephrasing this, the meaning is unclear.

Lines 406-410: It is too bad that the ending statement of the discussion pertains to a description of statistical analyses rather than a final concluding statement.

We’ve now revised the discussion to highlight why results are interesting while being honest and transparent about study limitations.

Methods

Line 484: Please consider double checking that the citation is correct.

Line 490: Please consider not starting a paragraph with “N.B.”.

Now corrected.

Lines 500-503: Please rephrase this, there might be a word missing as the overall meaning is not clear.

Line 506: Did the procedure include corrections specific to model fitting to account for the issue of overfitting? Was the sample group (training) specific to each treatment group or obtained from an average of all samples?

Yes, we implemented two concrete strategies to make sure our results are not due to overfitting. These are described, respectively, in steps 2 and 4 of the data-driven model description in the "EEG LZc models" section of the Methods document.

First, the correlations and R^2 values we report are all calculated based on predictions for held-out subjects (via leave-one-out cross-validation), ensuring that each model is trained and tested on different datapoints. Second, statistical significance (i.e. p-values) are estimated by repeating the whole procedure on randomly shuffled data. If the accuracy of the predictions were due to overfitting alone, the resulting R^2 values in the real and shuffled dataset would be identical. Since this is not the case, we can conclude that our predictions are statistically significant and not an artefact of overfitting.

Line 573: If two subjects are excluded from the analysis, why is the final $N=23$ and not 26? Is the initial N not 28?

It is two subjects from an MRI completing sample of 25. Three participants failed to complete the final MRI due to the Covid-19 pandemic. We now provide a table (final table) in the supplement listing exclusions and reasons for the exclusions.

Line 593: Please consider explaining or briefly mentioning these proper hypotheses. Are they based on previous studies or personal discussions?

A combination of both.

Lines 602-603: Is scan 2 compared to scan 3 directly? If so, is there a concern that the comparison is not between treatment and baseline but rather between high and low dose of drug?

That particular contrast was considered the most salient and of interest, as it's active drug versus placebo. 1mg psilocybin was verified to be inactive (EEG) but was always intended to be a placebo control.

Or is this a comparison of scan 2 (fitted to scan 1) and scan 3 (fitted to scan 1)?

The repeated measures ANOVAs addressed this. This is a test across all 3 timepoints. Otherwise, we held a special interest in the 1mg versus 25mg contrast as this was placebo versus active.

Line 675: (Formula) Does the I need to be written as lw ?

I'm sorry, it's very hard to know to what you're referring.

Line 734: "Chose" may be a typo for "choose".

Thank you for spotting this! It was indeed an error. Now corrected.

A sincere thank you for such a thorough review. It will help us improve this work considerably.

Robin Carhart-Harris, Ph.D.
Ralph Metzner Distinguished Professor | Neurology and Psychiatry
Weill Institute for Neurosciences
University of California, San Francisco

Sandler Neurosciences Building
675 Nelson Rising Ln.
San Francisco, CA 94158
Phone: (415) 476-2164
robin.carhart-harris@ucsf.edu
<https://www.carhartharrislab.com>

UNIVERSITY OF CALIFORNIA, SAN FRANCISCO

BERKELEY • DAVIS • IRVINE • LOS ANGELES • RIVERSIDE • SAN DIEGO • SAN FRANCISCO

SANTA BARBARA • SANTA CRUZ

Below are some final responses to the one remaining reviewer (reviewer 2) with comments.

Reviewer #2 (Remarks to the Author):

Overall, the responses to reviewer comments are not always constructive.

If one is suggesting including some information in the abstract, it is not appropriate that the authors simply answer that there is a word limit and they will not do it.

The exploratory nature of the study is now clearly labelled within the text – however, this should also be mentioned in the Abstract already.

The abstract now contains language that highlights the exploratory nature of the study.

Overall, I believe that both reviewers agreed that one is getting lost in the amount of analyses and how the authors are dealing with multiple comparisons.

We are addressing one reviewer now as the others were satisfied with our responses. We have formally shown how the outcomes interrelate (e.g., Fig 3) and have detailed how we corrected for multiple comparisons.

I overall come to the conclusion that this manuscript is overloaded for the given format. It is not possible to tie together all of these test and analyses and it does not get better with review rounds.

So for me it is overloaded and ultimately it is not understandable how all of these results belong together. A so long supplement with more and more analyses does not help to understand which are simply “additional nice to have analyses” and which of the analyses support the main claim of the article.

I think it is ok to do this – but for me it is not clear enough for a format with such high visibility, considering that the overall study was exploratory in its nature and not preregistered.

Other points:

I do not understand the response of the authors to Reviewer #1 requests about double-dipping. This requires more clarification why their procedures would not be double-dipping.

We have fully addressed this issue. Reviewer #1 was satisfied with our responses.

The responses to reviewer#1 request if an ANOVA is appropriate to test for robustness, is surprising – the authors might want to suggest alternative analyses.

As above.

The supplement is still having figures with bizarre bad resolution (e.g. S9), which require correction – I believe supplements are typically submitted as PDF – so this is the job of the authors.

The resolution of S9 is high. It is pasted below for ease of reference.

323
324

4.8. Figure S9. Bilateral amygdala BOLD RSFC analysis.

325
326
327
328
329
330

Figure S9. A) The bilateral amygdala is shown in magenta, from the Harvard-Oxford atlas, >50% threshold. Hot colours show the cluster-corrected map (threshold $Z = 2.3$, $p < 0.05$) for amygdala RSFC contrast, one-month post-25mg vs one-month post-1mg. Increased amygdala coupling can be seen with regions that overlap the so-called 'default-mode network'. Values from this timepoint-specific contrast, and the clusters therein, contributed to the amg-

Line 351:of the psychedelic trip“ should be reformulated to something like “induced subjective experienced altered states effects...” or some other formulation which points to the fact that it is about subjective experience, and avoiding the unscientific term “trip”.

This has now been revised throughout.

Related: I would encourage the authors to discuss more that there was only the measure of “intensity of drug effects” assessed. Which is a shame. Please verify that indeed there were no other measures taken for different types of subjective experiences. You might refer to other studies which do this with much more detail and explain why you did not do this, and discuss this limitation. Mainly: That one cannot say which aspect of the experience was relevant, as “intensity” is not really informative.

We accept this point and have now added a comment that other dimensions of subjective experience could be assessed. This sentence is included in the discussion as a limitation of the present work. Pasted below for ease of reference.

429 results imply that human brain changes as early as one-hour into a 25mg psilocybin experience—
430 and that seem closely related to the subjective psychedelic experience— can predict mental
431 health improvements one-month later. This finding therefore lends further, strong support⁴⁰ to the
432 position that the psychedelic experience is critically involved in the therapeutic effects of
433 psychedelic compounds. In terms of subjective effects, the present work only reports on generic
434 intensity. Future work could examine additional, more specific dimensions of experience, such as
435 emotional breakthrough. [REF]
436

Thank you for your review,

Robin Carhart-Harris and Colleagues

Manuscript Review: Human brain changes after first psilocybin use

The authors present a multimodal analysis of the pre- and post-treatment effects of a single dose of psilocybin in psychedelically-naïve human subjects (N=28) on measures of EEG, fMRI and DTI using a placebo-controlled design and a repeated measures analysis. The 25mg dose induced a significant decrease in axial diffusivity in prefrontal-subcortical tracts that corresponded with decreased modularity when assessing the pre- and post-effects. Initial increased cortical signal entropy response at 1-2 hours predicted an overall increase in psychological well-being one month later.

Major Strengths and Innovation

- The authors build on previous identification of unique effects that psilocybin, a 5-HT_{2A}R agonist, has on the overall modularity and entropy of the whole brain to predict how acute brain entropy can predict psychological benefit up to a month post dosing.
- The authors use a functionally inactive placebo (1mg dose of psilocybin) as well as a crossover design to allow for validation of psychological, structural and functional brain changes within each participant while controlling for the issue of finding a psychedelic placebo.
- The authors use a combination of multimodal imaging techniques which is essential to obtain a well-rounded perspective of the varying degrees of change that occur in the brain after psilocybin ingestion, especially with regards to region specific changes, and acute vs longer lasting effects.
- This study uses a multi-modal imaging approach to better quantify the enduring effect of psilocybin after a single dose. This study not only demonstrates that acute entropy can predict specific psychological outcomes, but that psilocybin has an enduring effect on DTI measures of white matter diffusivity and anisotropy and white matter tractography with region specificity.

Major Weaknesses

- The authors present a varied assortment of analyses and methods that do not always follow a logical order, especially in the context of how they relate to the overall hypothesis. A significant discussion and union of the analyses and their unified interpretation is also absent. Much of the data analysis seems to be exploratory rather than chosen with respect to an initial hypothesis. The paper seems exploratory to the point of running superfluous statistical analyses to find significant relationships. The paper clearly references previous studies and similar work in the field, yet the main findings are relatively adjacent to these previous findings and do not significantly advance the state-of-the-art as a reader might expect to see in this journal.
- This work is original and complements current minor gaps in the field of psychedelics in the context of neuroplasticity. However, the many different forms of statistical analyses paired with an unclear initial and end goal reduces the significance of this work and creates a constellation of findings that does not allow a significant impact.
- The conclusion and claims are supported by the various methods and analyses undertaken in the paper, although additional information relative to methodology may be needed to clarify why or how the underlying hypothesis was chosen. References are made to previous studies, but there is not enough discussion of how the new results affect the current field, which gaps are being challenged, and what the next steps in determining a new hypothesis could be.
- It may be worthwhile to provide additional information about how exploratory this work was meant to be. There are places throughout the manuscript where it is unclear how the data analysis technique is chosen (Lines 371-373, 401-402). It may be useful to follow a similar format per section (fMRI vs DTI vs psychological assessment) to clearly re-state the data analysis steps and then conclude on the main finding. Overall, the interpretation of results is unclear and disordered, which points to a need to review both the introduction and discussion sections of the paper. The overall concluding statement relates to statistical choices rather than an overarching theme, or final major finding(s). This lack of structure and clear scientific method reduces the impact of the results.
- The methodology does not seem to follow a direction with a clear hypothesis and points to the study as being primarily explorative, especially considering the many different types of analyses performed. The paper has interesting perspectives, with the findings relative to fMRI, DTI and LZc and how it relates to psychological outcomes at one-month post-drug administration, yet they are minor and do not advance the field as much as they could.
- While the methodology and general analysis may not be as clear, the Methods are well detailed and contain enough detail that this work could be reproduced.

Statistical Analysis

- The authors utilized various stringent statistical analyses across data types, including paired t-test for EEG data, a combination of using residuals and correlation to demonstrate the relationship between LZc and psychological outcomes, a mixed-effects GLM to compare the effect of treatment in rs-fMRI and emotional paradigm scan sessions, a bivariate regression to consider the interaction of psychophysiological effects, permutations to analyze the rs-fMRI matrix, and a repeated-measures ANOVA with TBSS to study the change between treatment groups.

Line-specific corrections:

General comments:

Throughout the paper, there are references that appear as a placeholder for a complete sentence. It would be useful to insert references at the end of a full sentence to improve the flow of the writing.

Introduction

Line 28: Please double check the formula for psilocybin.

Line 32: Throughout the manuscript, the references follow an incomplete sentence. Please consider reviewing proper citation methods and formatting.

Line 36: The phrasing of this sentence implies that the LSD study involved depressed patients. Please consider rewriting this.

Lines 38-39: Citation format

Line 41: Please define the knowledge gaps and how this relates to the hypothesis of the research study.

Line 45: Please clarify if this is rs-EEG or not.

Results

Line 65: Please consider explaining LZc in more detail here.

Line 82: Which ANOVA is used? This changes from ANOVA to inclusive ANOVA to repeated measures ANOVA to one-way ANOVA and is unclear.

Line 93: What feedback has been received? Is this based on result outcomes?

Line 122: Inclusive ANOVA- does this refer to a repeated measures ANOVA?

Line 155: Please consider explaining a bit more what modularity means.

Line 158: Please consider explicitly mentioning the figure within the Supplemental file.

Lines 189-190: Please consider explaining what IDED and EDS means, potentially introducing it before it appears in the figure legend.

Line 267: How was this normalized?

Lines 271-272: This is not a complete sentence.

Discussion

Line 301: Please consider reintroducing the overall experimental design and/or hypothesis driven question.

Lines 303-305: It is unclear if this sentence is necessary to this paragraph.

Lines 313-316: Please consider rephrasing this, the meaning is not as clear as it could be.

Line 321: Please consider explaining why DTI measures of AD are chosen based on the previous literature describing spine formation in the frontal cortex/ synaptic density changes and why AD is chosen over other DTI measures. It may be judicious to avoid overinterpretation of AD as it is a minor DTI metric and the biological interpretation is unclear to this day.

Line 338: This is the first mention of FA that seems to appear, why is this measure not discussed in more detail.

Lines 339-340: Please consider rephrasing "under-myelinated axons". This does not seem to be the correct terminology.

Line 362: This is not a proper citation.

Lines 375-378: Please consider rephrasing this sentence.

Line 383: This is not a complete sentence.

Line 391: This section follows a section discussing fMRI. This could be moved to the beginning of the discussion, otherwise as is, this reads like the beginning of another paper. Please consider changing this to flow better.

Lines 401-402: Please consider rephrasing this, the meaning is unclear.

Lines 406-410: It is too bad that the ending statement of the discussion pertains to a description of statistical analyses rather than a final concluding statement.

Methods

Line 484: Please consider double checking that the citation is correct.

Line 490: Please consider not starting a paragraph with "N.B."

Lines 500-503: Please rephrase this, there might be a word missing as the overall meaning is not clear.

Line 506: Did the procedure include corrections specific to model fitting to account for the issue of overfitting? Was the sample group (training) specific to each treatment group or obtained from an average of all samples?

Line 573: If two subjects are excluded from the analysis, why is the final N=23 and not 26? Is the initial N not 28?

Line 593: Please consider explaining or briefly mentioning these proper hypotheses. Are they based on previous studies or personal discussions?

Lines 602-603: Is scan 2 compared to scan 3 directly? If so, is there a concern that the comparison is not between treatment and baseline but rather between high and low dose of drug? Or is this a comparison of scan 2 (fitted to scan 1) and scan 3 (fitted to scan 1)?

Line 675: (Formula) Does the / need to be written as m ?

Line 734: "Chose" may be a typo for "choose".